# Exploration Bonus for Regret Minimization in Discrete and Continuous Average Reward MDPs

**Jian Qian, Ronan Fruit**
Sequel Team - Inria Lille
jian.qian@ens.fr, ronan.fruit@inria.fr

**Matteo Pirotta, Alessandro Lazaric**
Facebook AI Research
{pirotta, lazaric}@fb.com

## Abstract

The exploration bonus is an effective approach to manage the exploration-exploitation trade-off in Markov Decision Processes (MDPs). While it has been analyzed in infinite-horizon discounted and finite-horizon problems, we focus on designing and analysing the exploration bonus in the more challenging infinite-horizon undiscounted setting. We first introduce SCAL⁺, a variant of SCAL [1], that uses a suitable exploration bonus to solve any discrete *unknown weakly-communicating* MDP for which an upper bound $c$ on the span of the *optimal bias function* is known. We prove that SCAL⁺ enjoys the same regret guarantees as SCAL, which relies on the less efficient extended value iteration approach. Furthermore, we leverage the flexibility provided by the exploration bonus scheme to generalize SCAL⁺ to smooth MDPs with continuous state space and discrete actions. We show that the resulting algorithm (SCCAL⁺) achieves the same regret bound as UCCRL [2] while being the first implementable algorithm for this setting.

## 1 Introduction

While learning in an unknown environment, a reinforcement learning (RL) agent must trade off the *exploration* needed to collect information about the dynamics and reward, and the *exploitation* of the experience gathered so far to gain reward. An effective strategy to trade off exploration and exploitation is the *optimism in the face of uncertainty* (OFU) principle. A popular technique to ensure optimism is to use an *exploration bonus*. This approach has been successfully implemented in *H-step finite-horizon* and *infinite-horizon $\gamma$-discounted* settings with provable guarantees in finite MDPs. Furthermore, its simple structure (i.e., it only requires solving an estimated MDP with a reward increased by the bonus) allowed it to be integrated in deep RL algorithms [e.g., 3, 4, 5, 6]. As the exploration bonus is designed to bound estimation errors on the value function, it requires knowing the maximum reward $r_{\max}$ and the *intrinsic horizon* of the problem [e.g., 7, 8, 9] (e.g., $H$ in finite-horizon and $1/(1-\gamma)$ in discounted problems). Here we consider the challenging infinite-horizon undiscounted setting [10, Chap. 8], which generalizes the two previous settings when $H \to \infty$ and $\gamma \to 1$. While several algorithms implementing the OFU principle in this setting have been proposed [11, 2, 12, 1, 13], none of them exploits the idea of an exploration bonus.

In this paper we study the problem of defining and analysing an exploration bonus approach in the infinite-horizon undiscounted setting. Contrary to the other settings, in average reward there is no information about the intrinsic horizon. As a consequence, we follow the approach in [14, 1] and we assume that an upper-bound $c$ on the range of the optimal bias (i.e., value function) is known. We define SCAL⁺ and we show that its regret is bounded by $\widetilde{O}\big(\max\{c, r_{\max}\}\sqrt{\Gamma SAT}\big)$ w.h.p. for any MDP with $S$ states, $A$ actions and $\Gamma$ possible next states. We prove that the bonus used by SCAL⁺ ensures optimism using a novel technical argument. We no longer use an inclusion argument (i.e., the true MDP is contained in a set of plausible MDPs) but we reason directly at the level of the Bellman operator. We show that the optimistic Bellman operator defined by the empirical MDP with optimistic reward $\widehat{r}(s,a) + b(s,a)$ dominates the Bellman operator of the true MDP when *applied to*

*the optimal bias function*. This is sufficient to prove that the solution of the optimistic MDP is indeed (gain-)optimistic. This proof technique has two main advantages w.r.t. the inclusion argument. First, it directly applies to slightly perturbed empirical MDPs, without re-deriving confidence sets. Second, as we study the optimistic Bellman operator applied only to the optimal bias function (rather than all possible vector in $\mathbb{R}^S$), we save a factor $\sqrt{\Gamma}$ in designing the exploration bonus, compared to the (implicit) bounds obtained by algorithms relying on confidence sets on the MDP. Furthermore, as SCAL$^+$ only solves the estimated MDP with optimistic reward, it is computationally cheaper than UCRL-based algorithms, which require computing the optimal policy for an *extended* MDP with a continuous action space defined by the confidence set over MDPs.

Surprisingly, the "tighter" optimism of SCAL$^+$ does not translate into a better regret, which actually matches the one of SCAL and still depends on the factor $\sqrt{\Gamma}$. We isolate and discuss where the term $\sqrt{\Gamma}$ appears in the proof sketch of Sect. 3.3. While Azar et al. [8], Jin et al. [9] managed to remove the $\sqrt{\Gamma}$ term in the finite-horizon setting, their proof techniques cannot be directly applied to the infinite-horizon case. Recently Ortner [15] derived an algorithm achieving $O\left(\sqrt{t_{\mathrm{mix}} S A T}\right)$ regret bound under the assumption that the true MDP is *ergodic* ($t_{\mathrm{mix}}$ denotes the maximum *mixing time* of any policy). It remains an open question if a regret scaling with $\sqrt{S}$ (instead of $\sqrt{\Gamma S}$) can be achieved in the infinite-horizon case without any ergodicity assumption. We report preliminary experiments showing that the exploration bonus may indeed limit *over-exploration* and lead to better empirical performance w.r.t. approaches based on confidence intervals on the MDP itself (i.e., UCRL and SCAL). A more detailed comparison to existing literature is postponed to App. A.

To further illustrate the generality of the exploration bonus approach, we also present SCCAL$^+$, an extension of SCAL$^+$ to continuous state MDPs. As in [2, 16], we require the reward and transition functions to be Hölder continuous with parameters $\rho_L$ and $\alpha$. SCCAL$^+$ is also the first implementable algorithm in continuous average reward problems with theoretical guarantees (existing algorithms with theoretical guarantees such as UCCRL [2] cannot be implemented). The key result is a regret bound of $\widetilde{O}\left(\max\{c, r_{\max}\}\rho_L \sqrt{A} T^{(\alpha + 2)/(2\alpha + 2)}\right)$ w.h.p. Finally, we provide an empirical comparison of SCCAL$^+$ with a Q-learning algorithm with exploration bonus for average reward problems (RVIQ-UCB) inspired by [17, 9] and the results in this paper (to deal with continuous states).

## 2 Preliminaries

We consider a *weakly-communicating* MDP [10, Sec. 8.3] $M = (\mathcal{S}, \mathcal{A}, p, r)$ with state space $\mathcal{S}$ and action space $\mathcal{A}$. Every state-action pair $(s, a)$ is characterized by a reward distribution with mean $r(s, a)$ and support in $[0, r_{\max}]$, and a transition distribution $p(\cdot|s, a)$ over next states. In this section, we assume the finite case (i.e., $|\mathcal{S}|, |\mathcal{A}| < +\infty$), although all following definitions extend to continuous state spaces under mild assumptions on $r$ and $p$ (see Sect. 4). We denote by $S = |\mathcal{S}|$ and $A = |\mathcal{A}|$ the number of states and action, by $\Gamma(s, a) = \|p(\cdot|s, a)\|_0$ the number of states reachable by selecting action $a$ in state $s$, and by $\Gamma = \max_{s,a} \Gamma(s, a)$ its maximum. A stationary Markov randomized policy $\pi : \mathcal{S} \to P(\mathcal{A})$ maps states to distributions over actions. The set of stationary randomized (resp. deterministic) policies is denoted by $\Pi^{\mathrm{SR}}$ (resp. $\Pi^{\mathrm{SD}}$). Any policy $\pi \in \Pi^{\mathrm{SR}}$ has an associated *long-term average reward* (or gain) and a *bias function* defined as

$$g^\pi(s) := \lim_{T \to +\infty} \mathbb{E}_s^\pi\left[\frac{1}{T}\sum_{t=1}^T r(s_t, a_t)\right] \quad \text{and} \quad h^\pi(s) := \underset{T \to +\infty}{C\text{-}\lim} \mathbb{E}_s^\pi\left[\sum_{t=1}^T \big(r(s_t, a_t) - g^\pi(s_t)\big)\right],$$

where $\mathbb{E}_s^\pi$ denotes the expectation over trajectories generated starting from $s_1 = s$ with $a_t \sim \pi(s_t)$. The bias $h^\pi(s)$ measures the expected total difference between the reward and the stationary reward in *Cesaro-limit* (denoted by $C\text{-}\lim$). Accordingly, the difference of bias $h^\pi(s) - h^\pi(s')$ quantifies the (dis-)advantage of starting in state $s$ rather than $s'$. We denote by $sp(h^\pi) := \max_s h^\pi(s) - \min_s h^\pi(s)$ the *span* of the bias function. In weakly communicating MDPs, any optimal policy $\pi^* \in \arg\max_\pi g^\pi(s)$ has *constant* gain, i.e., $g^{\pi^*}(s) = g^*$ for all $s \in \mathcal{S}$. Moreover, there exists a policy $\pi^* \in \arg\max_\pi g^\pi(s)$ for which $(g^*, h^*) = (g^{\pi^*}, h^{\pi^*})$ satisfy the *optimality equation*,

$$\forall s \in \mathcal{S}, \qquad h^*(s) + g^* = Lh^*(s) := \max_{a \in \mathcal{A}}\{r(s, a) + p(\cdot|s, a)^\mathsf{T} h^*\}, \tag{1}$$

where $L$ is the *optimal* Bellman operator. Finally, $D = \max_{s \neq s'}\{\tau(s \to s')\}$ denotes the diameter of $M$, where $\tau(s \to s')$ is the minimal expected number of steps needed to reach $s'$ from $s$.

**Input:** Confidence $\delta \in ]0, 1[$, $r_{\max}$, $\mathcal{S}$ ($\mathcal{I}$ for SCCAL$^+$), $\mathcal{A}$, $c \geq 0$ (and $\rho_L$ and $\alpha$ for SCCAL$^+$)
**For** episodes $k = 1, 2, ...$ **do**
1. Set $t_k = t$ and episode counters $\nu_k(s, a) = 0$.
2. Compute estimates $\widehat{p}_k^+(I(s')|I(s), a)$, $\widehat{r}_k^+(I(s), a)$, $b_k(I(s), a)$ (Eq. 3 or 6) and build the MDP $\widehat{M}_k^+$
   (SCAL$^+$) or $\widehat{M}_k^{ag+}$ (SCCAL$^+$).
3. Compute an $\frac{r_{\max}}{t_k}$-approximate solution of Eq. 4 for $\widehat{M}_k^+$ (SCAL$^+$) or $\widehat{M}_k^{ag+}$ (SCCAL$^+$) using SCOPT
4. Sample action $a_t \sim \pi_k(\cdot|I(s_t))$.
5. **While** $\nu_k(I(s_t), a_t) < \max\{1, N_k(I(s_t), a_t)\}$ **do**
   (a) Execute $a_t$, obtain reward $r_t$, and observe next state $s_{t+1}$.
   (b) Increment counter $\nu_k(s_t, a_t) \mathrel{+}= 1$.
   (c) Sample action $a_{t+1} \sim \pi_k(\cdot|I(s_{t+1}))$ and increment $t \mathrel{+}= 1$.
6. Increment counters $N_{k+1}(s, a) := N_k(s, a) + \nu_k(s, a)$ for all $s, a$.

Figure 1: Shared pseudo-code for SCAL$^+$ and SCCAL$^+$. For SCAL$^+$ $I(s) = s$ by definition.

**Learning Problem.** Let $M^*$ be the true MDP. We consider the learning problem where $\mathcal{S}$, $\mathcal{A}$ and $r_{\max}$ are *known*, while rewards $r$ and dynamics $p$ are *unknown* and need to be estimated *on-line*. We evaluate the performance of a learning algorithm $\mathfrak{A}$ after $T$ time steps by its cumulative *regret* $\Delta(\mathfrak{A}, T) = \sum_{t=1}^T (g^* - r_t(s_t, a_t))$. Finally, we make the following assumption.

**Assumption 1.** *There exists a* known *upper-bound $c > 0$ to the optimal bias span i.e., $c \geq sp(h^*)$.*

This assumption is common in the literature [see e.g., 18, 2, 1]. Such a bound to the "range" of the value function is already available in discounted and finite horizon problems (i.e., as $\frac{1}{1-\gamma}$ and $H$), so Asm. 1 is not more restrictive. While the span $sp(h^*)$ is a non-trivial function of the dynamics and the rewards of the MDP, some intuition about how the cumulative reward varies depending on different starting states is often available. Furthermore, as $sp(h^*) \leq r_{\max} D$ [e.g., 14], it is sufficient to have prior knowledge about the diameter $D$ and the range of the reward $r_{\max}$, to provide a rough upper-bound on the span.

## 3 SCAL$^+$: SCAL with exploration bonus

In this section, we introduce SCAL$^+$, the first online RL algorithm –in the infinite horizon undiscounted setting– that leverages an *exploration bonus* to achieve *near-optimal* regret guarantees. Similar to SCAL [1], SCAL$^+$ takes as input an upper-bound $c$ on the optimal bias span (i.e., $sp(h^*) \leq c$) to constrain the planning problem solved over time. The crucial difference is that SCAL$^+$ does not compute an *optimistic* MDP within a high-probability confidence set, but it directly computes the optimal policy of the *estimated* MDP, with the reward increased by an *exploration bonus*. The bonus is carefully tuned so as to guarantee optimism and small regret at the same time (Thm. 1).

### 3.1 The Algorithm

Similar to other OFU-based algorithms, SCAL$^+$ proceeds in episodes (see Fig. 1)[1]. Denote by $t_k$ the starting time of episode $k$, $N_k(s, a, s')$ the number of observations of tuple $(s, a, s')$ before episode $k$ and $N_k(s, a) := \sum_{s'} N_k(s, a, s')$. We define the estimators of the transitions and rewards as

$$\widehat{p}_k^+(s'|s, a) = \frac{N_k(s, a, s')}{N_k(s, a) + 1} + \frac{\mathbb{1}(s' = \overline{s})}{N_k(s, a) + 1}, \quad \overline{r}_k(s, a) = \sum_{t=1}^{t_k-1} \frac{r_t(s_t, a_t) \mathbb{1}\big((s_t, a_t) = (s, a)\big)}{N_k(s, a)} \quad (2)$$

where $\overline{s} \in \mathcal{S}$ is an arbitrary state and $\overline{r}_k(s, a) := r_{\max}$, $\widehat{p}_k^+(s'|s, a) = 1/S$ when $N_k(s, a) = 0$. The transition model $\widehat{p}_k^+(s'|s, a)$ is a biased (but asymptotically consistent) estimator of $p(s'|s, a)$. We further define the exploration bonus

$$b_k(s, a) := (c + r_{\max}) \underbrace{\sqrt{\frac{\ln\big(20SAN_k^+(s, a)/\delta\big)}{N_k^+(s, a)}}}_{:=\beta_k^{sa}} + \frac{c}{N_k(s, a) + 1}, \quad (3)$$

where $N_k^+(s,a) = \max\{1, N_k(s,a)\}$. Intuitively, the exploration bonus is large for poorly visited state-action pairs, while it decreases with the number of visits. A crucial aspect in the formulation of $b_k$ is that it scales with the span $c$. In fact, the exploration bonus is not used to obtain an upper-confidence bound on the reward (setting $b_k(s,a) = \beta_k^{sa}$ would be sufficient), but it is designed to take into consideration how estimation errors on $\overline{p}$ and $\overline{r}$, which are bounded by $\beta_k^{sa}$, may propagate to the bias and gain through repeated applications of the Bellman operator. As the span $c$ provides prior knowledge about the "range" of the optimal bias function, the exploration bonus is obtained by considering that "local" estimation errors may be amplified *up to* a factor $c$. The specific shape of $b_k$ and $\beta_k^{sa}$ and their theoretical properties are derived in Lem. 1. At each episode $k$, SCAL$^+$ builds an MDP $\widehat{M}_k^+ = (\mathcal{S}, \mathcal{A}^+, \widehat{p}_k^+, \widehat{r}_k^+)$ obtained by duplicating every action in $\mathcal{A}$ with transition probabilities unchanged and optimistic reward set to 0. Formally, let $\mathcal{A}^+ := \mathcal{A} \times \{1,2\}$ and we denote any pair $(a,i) \in \mathcal{A} \times \{1,2\}$ by $a_i$. We then define $\widehat{r}_k^+(s,a_i) := \left(\overline{r}_k(s,a) + b_k(s,a)\right) \cdot \mathbb{1}(i = 1)$. SCAL$^+$ proceeds by computing the optimal policy of the MDP $\widehat{M}_k^+$ subject to the constrains on the bias span:

$$\pi_k := \arg \sup_{\pi \in \Pi_c(\widehat{M}_k^+)} \{g^\pi\}; \qquad g_c^*(\widehat{M}_k^+) := \sup_{\pi \in \Pi_c(\widehat{M}_k^+)} \{g^\pi\}, \qquad (4)$$

where the constraint set is $\Pi_c(M) := \left\{\pi \in \Pi^{\text{SR}} : sp\left(h^\pi\right) \leq c \wedge sp\left(g^\pi\right) = 0\right\}$. The optimal policy is executed until the number of visits in at least one state-action pair during the episode has doubled.

Problem 4 is well posed and can be solved using SCOPT. Let $\widehat{L}^+$ be the optimal Bellman operator associated to $\widehat{M}_k^+$, given $v \in \mathbb{R}^S$ and $c \geq 0$, we define the value operator $\widehat{T}_c^+ : \mathbb{R}^S \to \mathbb{R}^S$ as

$$\widehat{T}_c^+ v = \Gamma_c \widehat{L}^+ v = \begin{cases} \widehat{L}^+ v(s) & \forall s \in \left\{s \in \mathcal{S} | \widehat{L}^+ v(s) \leq \min_s \{\widehat{L}^+ v(s)\} + c\right\} \\ c + \min_s\{\widehat{L}^+ v(s)\} & \textit{otherwise} \end{cases} \quad (5)$$

where $\Gamma_c$ is the span constrain projection operator (see [1, App. D] for details). In other words, operator $\widehat{T}_c^+$ applies a *span truncation* to the one-step application of $\widehat{L}^+$, which guarantees that $sp(\widehat{T}_c^+ v) \leq c$. Given a vector $v_0 \in \mathbb{R}^S$ and a reference state $\overline{s}$, SCOPT runs relative value iteration where $\widehat{L}^+$ is replaced by $\widehat{T}_c^+$ as $v_{n+1} = \widehat{T}_c^+ v_n - \widehat{T}_c^+ v_n(\overline{s})e$. The policy $\pi_k$ returned by SCOPT takes action in the *augmented* set $\mathcal{A}^+$ and it can be "projected" on $\mathcal{A}$ as $\pi_k(s,a) \leftarrow \pi_k(s,a_1) + \pi_k(s,a_2)$ (we use the same notation for the two policies), which is the policy actually executed through the episode. Following similar steps as in [1], we can prove that $\widehat{M}_k^+$ satisfies all sufficient conditions for SCOPT to converge and return the optimal policy (see App. B).

**Proposition 1.** *The MDP $\widehat{M}_k^+$ satisfies the following properties: 1) the associated optimal Bellman operator $\widehat{L}^+$ is a $\gamma$-span-contraction; 2) all policies are unichain; 3) the operator $\widehat{T}_c^+$ is globally feasible at any vector $v \in \mathbb{R}^S$ such that $sp(v) \leq c$, i.e., for all $s \in \mathcal{S}$, $\min_{a \in \mathcal{A}}\{r(s,a) + p(\cdot|s,a)^\mathsf{T} v\} \leq \min_{s'}\{Lv(s')\} + c$. As a consequence, SCOPT converges and returns a policy $\pi_k$ solving* (4).

### 3.2 Optimistic Exploration Bonus

All regret proofs for OFU-based algorithms rely on the property that the optimal gain of the MDP used to compute $\pi_k$ ($\widehat{M}_k^+$ in our case) is an upper-bound on $g^*$. If we want to use the same proof technique for SCAL$^+$, we need to ensure that the policy $\pi_k$ is *gain-optimistic*, i.e., $\widehat{g}_k^+ := g_c^*(\widehat{M}_k^+) \geq g^*$.

Recall that the optimal gain and bias of the true MDP $(g^*, h^*)$ satisfy the optimality equation $Lh^* = h^* + g^*e$ where $e = (1, \ldots, 1)$. Since $sp(h^*) \leq c$ (by assumption), we also have $sp(Lh^*) = sp(h^* + g^*e) = sp(h^*) \leq c$ and so $T_c h^* = Lh^*$. A minor variation to Lemma 8 of Fruit et al. [1] shows that a sufficient condition to prove optimistic gain is to show that the operator $\widehat{T}_c^+$ is optimistic w.r.t. its exact version when applied to the optimal bias function, i.e., (see Prop. 3 in App. B)

$$\widehat{T}_c^+ h^* \geq h^* + g^*e = T_c h^*.$$

As the truncation operated by $T_c$ (i.e., $\Gamma_c$) is monotone, this inequality is implied by $\widehat{L}_k^+ h^* \geq Lh^*$. Finally, since $\widehat{p}_k^+(s'|s, a_1) = \widehat{p}_k^+(s'|s, a_2) = \widehat{p}_k(s'|s, a)$ and $\widehat{r}_k^+(s, a_2) \leq \widehat{r}_k^+(s, a_1)$ it is immediate to see that $\widehat{L}_k^+ h^* = \widehat{L}_k h^*$, thus implying that a sufficient condition for $\widehat{g}_k^+ \geq g^*$ is to have $\widehat{L}_k h^* \geq Lh^*$, which reduces to verifying optimism for the Bellman operator of $\widehat{M}_k^+$ when applied to the exact optimal bias function. The exploration bonus is tailored to achieve this condition with high probability.

**Lemma 1.** *Denote by $\widehat{L}_k$ the optimal Bellman operator of $\widehat{M}_k$. With probability at least $1 - \frac{\delta}{5}$, for all $k \geq 1$, $\widehat{L}_k h^* \geq L h^*$ (componentwise) and as a consequence $\widetilde{g}_k^+ \geq g^*$.*

*Proof (see App. D).* By using Hoeffding-Azuma inequality and union bound, we can show that for all $k \geq 1$, $|\overline{r}_k(s,a) - r(s,a)| \leq r_{\max} \beta_k^{sa}$ and $|(\overline{p}_k(\cdot|s,a) - p(\cdot|s,a))^\intercal h^*| \leq c\, \beta_k^{sa}$ w.h.p. ($\overline{p}_k$ is the MLE of $p$). We also need to take into account the small bias introduced by $\widehat{p}_k(\cdot|s,a)$ compared to $\overline{p}_k(\cdot|s,a)$ which is not bigger than $c/(N_k(s,a)+1)$ by definition. Then, with high probability, for all $k \geq 1$, $\overline{r}_k(s,a) + b_k(s,a) + \widehat{p}_k(\cdot|s,a)^\intercal h^* \geq r(s,a) + p(\cdot|s,a)^\intercal h^*$ for all $(s,a) \in \mathcal{S} \times \mathcal{A}$. $\qquad\square$

The argument used to prove optimism (Lem. 1)) significantly differs from the one used for UCRL and SCAL. Confidence-based methods compute the optimal policy of an *extended* MDP that *"contains"* the true MDP $M^*$ (w.h.p.), which directly implies that the gain of the extended MDP is bigger than $g^*$. The main advantage of our argument is that it allows for a *"tighter"* optimism (i.e., less prone to over-exploration). In fact, the exploration bonus quantifies by how much $\widehat{L}_k^+ h^*$ is bigger than $L h^*$ and it approximately scales as $b_k(s,a) = \widetilde{\Theta}\big(\max\{r_{\max}, c\}/\sqrt{N_k(s,a)}\big)$. In contrast, UCRL and SCAL use an optimistic Bellman operator $\widetilde{L}$ such that $\widetilde{L} h^*$ is bigger than $L h^*$ by respectively $\widetilde{\Theta}\big(r_{\max} D \sqrt{\Gamma/N_k(s,a)}\big)$ (UCRL) and $\widetilde{\Theta}\big(\max\{r_{\max}, c\}\sqrt{\Gamma/N_k(s,a)}\big)$ (SCAL). In other words, the optimism in SCAL⁺ is tighter by a multiplicative factor $\sqrt{\Gamma}$.

### 3.3   Regret Analysis of SCAL⁺

We report the main result of this section.

**Theorem 1.** *For any* weakly communicating *MDP $M$ such that $sp\,(h^*) \leq c$, with probability at least $1 - \delta$ it holds that for any $T \geq 1$, the regret of SCAL⁺ is bounded as*

$$\Delta(\text{SCAL}^+, T) = O\left( \max\{r_{\max}, c\} \left( \sqrt{\Big( \sum_{s,a} \Gamma(s,a) \Big) T \ln\,(T/\delta)} + S^2 A \ln^2 \Big( \frac{T}{\delta} \Big) \right) \right)$$

Since the optimism in SCAL⁺ is tighter than in UCRL and SCAL by a factor $\sqrt{\Gamma}$, one may expect to get a regret bound scaling as $c\sqrt{SAT}$ instead of $c\sqrt{\Gamma SAT}$, thus matching the lower bound of Jaksch et al. [11] as for the dependency in $S$. Unfortunately, such a bound seems difficult to achieve with SCAL⁺ (and even SCAL) due to the correlation between $h_k$ and $\overline{p}_k$ (see App. D). Azar et al. [8] managed to achieve the optimal dependence in $S$ in *finite-horizon problems*. In this setting, the definition of regret is different and it is not clear whether it is possible to adapt their guarantees and techniques to infinite horizon without introducing a $\Theta(T)$-term. Agrawal and Jia [19] showed the optimistic posterior sampling has a regret of $\widetilde{O}(D\sqrt{SAT})$ in the infinite horizon undiscounted setting. Unfortunately, their proof critically relies on the concentration inequality $|(\overline{p}_k(\cdot|s,a) - p(\cdot|s,a))^\intercal h_k| \lesssim r_{\max} D \beta_k^{sa}$ which is incorrect.[2] It remains as an open question whether the $\sqrt{\Gamma}$ term can be actually removed.

Finally, SCAL⁺'s regret does not scale $\min\{r_{\max}D, c\}$ as for SCAL, implying that SCAL⁺ may perform worse when $c$ is too large. The difference resides in the fact SCAL builds an extended MDP that contains the true MDP (w.h.p.). The shortest path between two states in the extended MDP is therefore shorter than in the true MDP and consequently, the diameter of the extended MDP is smaller than the true diameter $D$. This explains why the regret of SCAL depends on both $D$ and $c$ (which is provided as input to the algorithm). Unfortunately, in SCAL⁺ it is not clear how to bound the diameter of $\widehat{M}_k^+$ and the only information that can be exploited to bound the regret is the constraint $c$.

## 4   SCCAL⁺: SCAL⁺ **for continuous state space**

We now consider an MDP with continuous state space $\mathcal{S} = [0,1]$ and discrete action space $\mathcal{A}$. In general, it is impossible to learn an *arbitrary* real-valued function with only a *finite* number of samples. We therefore introduce the same smoothness assumption as Ortner and Ryabko [2]:

**Assumption 2** (Hölder continuity). *There exist $\rho_L, \alpha > 0$ s.t. for any two states $s, s' \in \mathcal{S}$ and any action $a \in \mathcal{A}$, $|r(s,a) - r(s',a)| \leq r_{\max}\rho_L|s - s'|^\alpha$  and  $\|p(\cdot|s,a) - p(\cdot|s',a)\|_1 \leq \rho_L|s - s'|^\alpha$.*

As in Sec. 3, we start by introducing our proposed algorithm SCCAL⁺ which is a variant of SCAL⁺ for continuous state space (Sec. 4.1), and then analyze its regret (Sec. 4.2).

## 4.1 The algorithm

In order to apply SCAL⁺ to a continuous problem, we discretize the state space as in [2]. We partition $\mathcal{S}$ into $S$ intervals defined as $I_1 := \left[0, \frac{1}{S}\right]$ and $I_k = \left]\frac{k-1}{S}, \frac{k}{S}\right]$ for $k = 2, \ldots, S$. The set of *aggregated* states is then $\mathcal{I} := \{I_1, \ldots, I_S\}$ ($|\mathcal{I}| = S$). The *number of intervals* $S$ is a parameter of the algorithm and plays a central role in its performance. Note that the terms $N_k(s, a, s')$ and $N_k(s, a)$ defined in Sec. 3 are still well-defined for $s$ and $s'$ lying in $[0, 1]$ but are 0 except for a finite number of $s$ and $s'$. For any subset $I \subseteq \mathcal{S}$, the sum $\sum_{s \in I} u_s$ is also well-defined as long as the collection $(u_s)_{s \in I}$ contains only a finite number of non-zero elements. We can therefore define the *aggregated* counts, rewards and transition probabilities for all $I, J \in \mathcal{I}$ as: $N_k(I, a) := \sum_{s \in I} N_k(s, a)$,

$$\overline{r}_k^{ag}(I, a) := \frac{1}{N_k(I, a)} \sum_{t=1}^{t_k - 1} r_t(s_t, a_t) \mathbb{1}(s_t \in I, a_t = a), \quad \overline{p}_k^{ag}(J|I, a) := \frac{\sum_{s' \in J} \sum_{s \in I} N_k(s, a, s')}{\sum_{s \in I} N_k(s, a)}.$$

Similar to Eq. 3, we define the exploration bonus of an aggregated state as

$$b_k(I, a) := (c + r_{\max}) \left(\beta_k^{Ia} + \rho_L S^{-\alpha}\right) + \frac{c}{N_k(I, a) + 1} \tag{6}$$

where $\beta_k^{Ia}$ is defined in (3). The main difference is an additional $O(c\rho_L S^{-\alpha})$ term that accounts for the fact that the states that we aggregate are not completely identical but have parameters that differ by at most $\rho_L S^{-\alpha}$. We pick an arbitrary reference aggregated state $\overline{I}$ and define $\widehat{M}_k^{ag} = (\mathcal{I}, \mathcal{A}, \widehat{p}_k^{ag}, \widehat{r}_k^{ag})$, the aggregated (discrete) analogue of $\widehat{M}_k$ defined in Sec. 3, where $\widehat{r}_k^{ag} = \overline{r}_k^{ag} + b_k$ and

$$\widehat{p}_k^{ag}(J|I, a) := \frac{N_k(I, a)\overline{p}_k^{ag}(J|I, a)}{N_k(I, a) + 1} + \frac{\mathbb{1}(J = \overline{I})}{N_k(I, a) + 1},$$

Similarly we "augment" $\widehat{M}_k^{ag}$ into $\widehat{M}_k^{ag+} = (\mathcal{I}, \mathcal{A}^+, \widehat{p}_k^{ag+}, \widehat{r}_k^{ag+})$ (analogue of $\widehat{M}_k^+$ in Sec. 3) by duplicating each transition in $\widehat{M}_k^{ag}$. At each episode $k$, SCCAL⁺ uses SCOPT (with the same parameters as in Sec. 3) to solve optimization problem (4) on $\widehat{M}_k^{ag+}$. This is possible because although the state space of $M^*$ is *uncountable*, $\widehat{M}_k^{ag+}$ has only $S < +\infty$ states. SCOPT returns an *optimistic* optimal policy $\pi_k$ satisfying the span constraint. This policy is defined in the discrete *aggregated* state space but can easily be extended to the continuous case by setting $\pi_k(s, a) := \pi_k(I(s), a)$ for any $(s, a)$ (with $I(s)$ mapping a state to the interval containing it).

## 4.2 Regret Analysis of SCCAL⁺

This section is devoted to the regret analysis of SCCAL⁺, with the main result summarized in Thm.2.

**Theorem 2.** *For any MDP $M$ satisfying Asm. 2 and such that $sp\left(h_M^*\right) \leq c$, with probability at least $1 - \delta$ it holds that for any $T \geq 1$, the regret of SCCAL⁺ is bounded as*

$$\Delta(\text{SCCAL}^+, T) = O\left(\max\{r_{\max}, c\}\left(S\sqrt{AT\ln(T/\delta)} + S^2 A \ln^2(T/\delta) + \rho_L S^{-\alpha} T\right)\right)$$

*By setting $S = \left(\alpha \rho_L \sqrt{\frac{T}{A}}\right)^{1/(\alpha+1)}$ the bound becomes: $\widetilde{O}\left(\max\{r_{\max}, c\}\rho_L^{\frac{1}{(\alpha+1)}} A^{\frac{\alpha}{(2\alpha+2)}} T^{\frac{(\alpha+2)}{(2\alpha+2)}}\right)$.*

Thm. 2 shows that SCCAL⁺ achieves the same regret as UCCRL [2] while being the only *implementable* algorithm with such *theoretical guarantees* for this setting. Thm. 2 can be extended to the more general case where $\mathcal{S}$ is $d$-dimensional. As pointed out by [2], in this case $S^d$ intervals are needed for the discretization leading to a regret bound of order $\widetilde{O}(T^{(2d+\alpha)/(2d+2\alpha)})$ after tuning $S = T^{1/(2d+2\alpha)}$. Finally, we believe that SCCAL⁺ can be extended to the setting considered by [16] where, in addition to Hölder conditions, the transition function is assumed to be $\kappa$-times smoothly differentiable. In the case of Lipschitz model, i.e., $\alpha = 1$, this means that it is possible obtain an asymptotic regret (as $\kappa \to \infty$) of $\widetilde{O}(T^{2/3})$ while SCCAL⁺ is achieving $\widetilde{O}(T^{3/4})$.

**Proof sketch.** Thm. 2 can be seen as a generalization of Thm. 1 but the continuous nature of the state space makes the analysis more difficult. The *main technical challenge* lies in relating two MDPs with *different state spaces*: $\widehat{M}_k^{ag}$ (with *finite* state space) and $M^*$ (with continuous state space). For instance, It is necessary to compare these two MDPs to prove *optimism*. To facilitate the comparison, we introduce an "intermediate" MDP $\widehat{M}_k$ which has *continuous* state space like $M^*$, but which also depends on the samples collected before episode $k$ like $\widehat{M}_k^{ag}$.

**Definition 1** (Empirical MDP with continuous state space). *Let $\widehat{M}_k = (\mathcal{S}, \mathcal{A}, \widehat{p}_k, \widehat{r}_k)$ be the continuous state space MDP s.t. for all $(s,a) \in \mathcal{S} \times \mathcal{A}$, $\overline{r}_k(s,a) := \overline{r}_k^{ag}(I(s),a)$,*

$$\widehat{r}_k(s,a) := \overline{r}_k(s,a) + b_k(I(s),a) \;\; and \;\; \widehat{p}_k(s'|s,a) := \frac{N_k(I(s),a)\overline{p}_k(s'|s,a)}{N_k(I(s),a)+1} + \frac{S \cdot \mathbb{1}(s' \in I(\overline{s}))}{N_k(I(s),a)+1}$$

*where $I : \mathcal{S} \to \mathcal{I}$ is the function mapping a state $s$ to the interval containing $s$, and $\overline{p}_k(s'|s,a)$ is the Radon-Nikodym derivative of the cumulative density function $F(s) = \sum_{s' \leq s} \frac{\sum_{x \in I(s)} N_k(x,a,s')}{N_k(I(s),a)}$.*

MDP $\widehat{M}_k$ is designed so that: 1) the reward function is *piece-wise* constant over any interval in $\mathcal{I}$ and matches the reward function of $\widehat{M}_k^{ag}$, 2) the transitions integrated over $s' \in J \in \mathcal{I}$ are piece-wise constant and match the transitions of the discrete state space MDP $\widehat{M}_k^{ag}$. More precisely, $\forall J \in \mathcal{I}$, $\int_J \overline{p}_k(s'|s,a)\mathrm{d}s' = \overline{p}_k^{ag}(J|I(s),a)$ and so $\forall (s,J) \in \mathcal{S} \times \mathcal{I}$:

$$\int_J \widehat{p}_k(s'|s,a)\mathrm{d}s' = \frac{N_k(I(s),a)\overline{p}_k^{ag}(J|I(s),a)}{N_k(I(s),a)+1} + \frac{S \int_J \mathbb{1}(s' \in I(\overline{s}))\mathrm{d}s'}{N_k(I(s),a)+1} = \widehat{p}_k^{ag}(J|I(s),a) \quad (7)$$

This ensures that $\widehat{M}_k^{ag}$ and $\widehat{M}_k$ can be easily compared (and as a consequence, so can $\widehat{M}_k^{ag+}$ and $\widehat{M}_k^+$, the augmented versions of $\widehat{M}_k^{ag}$ and $\widehat{M}_k$) although they have *different state spaces* and obtain:

**Lemma 2.** *For any $k \geq 1$, $\widehat{g}_k^{ag+} := g_c^*(\widehat{M}_k^{ag+}) = \widehat{g}_k^+ := g_c^*(\widehat{M}_k^+)$*

*Proof (see App. C.2).* We notice that for any continuous function $v(s)$ defined on $\mathcal{S}$ and piece-wise constant on the intervals of $\mathcal{I}$, we can associate a discrete function $v'(I)$ (defined on $\mathcal{I}$) such that for all $s \in \mathcal{S}$, $v'(I(s)) = v(s)$. Let $v_0 = 0$ (continuous function) and denote by $v_0'$ its discrete analogue. We define the sequences $(v_n)_{n \in \mathbb{N}}$ and $(u_n)_{n \in \mathbb{N}}$ by recursively applying $\widehat{T}_c^+$ and $\widehat{T}_c^{ag+}$ respectively: $v_{n+1} := \widehat{T}_c^+ v_n$ and $u_{n+1} := \widehat{T}_c^{ag+} u_n$ with $u_0 := v_0'$. It is easy to show that for all $n$, $v_n$ is piece-wise constant and its discrete analogue is $u_n$ i.e., $u_n = v_n'$. Therefore the sequences $v_{n+1}(s) - v_n(s)$ and $u_{n+1}(I(s)) - u_n(I(s))$ have the same limits, respectively $\widehat{g}_k^+$ and $\widehat{g}_k^{ag+}$. $\quad\square$

Leveraging Lem. 2, it is sufficient to compare the gains of $\widehat{M}_k^+$ and $M^*$ to prove optimism. Since both MDPs have the same (continuous) state space, we can proceed as in Sec. 3.2 and just show that $\widehat{L}_k h^* \geq L h^*$ (analogue of Lem. 1), with the difference that $h^*$ is defined on a continuous space.

**Lemma 3.** *Denote by $\widehat{L}_k$ the optimal Bellman operator of $\widehat{M}_k$. With probability at least $1 - \frac{\delta}{5}$, for all $k \geq 1$ we have $\widehat{L}_k h^* \geq L h^*$ (on the whole state space) and as a consequence $\widehat{g}_k^{ag+} \geq g^*$.*

*Proof (see Lem. 4 and 5) in App. C).* The proof is similar to Lem. 1: we compare $\widehat{r}_k$ and $\widehat{p}_k$ with the true reward function $r$ and transition probabilities $p$ using *concentration inequalities*. Due to the aggregation of states, there are two major differences with the discrete case. The first difference is that $\widehat{p}_k$ is even more *biased* than before. Thanks to the smoothness assumption (Asm. 2), the *extra bias* is only of order $O(LS^{-\alpha})$ (this explains why this term appears in the definition of the bonus in (6)). The second difference is that since there are *uncountably many states*, it is impossible to use a *union bound argument* on the set of states (like in Lem. 1). Instead, we show using *optional skipping* that the terms of interest are *martingales* and we apply Azuma's and Freedman's inequalities. $\quad\square$

The rest of the proof is similar to SCAL$^+$ with additional steps to deal with the continuous state space.

## 5 Numerical Simulations

We design experiments to investigate the learning performance in discrete and continuous MDP (see App. E for details). In the **discrete** case, the main theoretical open question is whether the tighter exploration bonus does translate in a better regret, that is, whether the dependency on the branching factor $\Gamma$ in the regret bound is due to the analysis or not. Unfortunately, it is difficult to design experiments to thoroughly investigate the actual dependency. First, it is challenging to design MDPs with all parameters fixed (i.e., gain, span, diameters, number of states and actions) but $\Gamma$ (e.g., the bigger $\Gamma$, the smaller the span as the MDP is more connected). Furthermore, the regret bound is worst-case w.r.t. all MDPs with a given set of parameters, which is difficult to design in practice. For these reasons, instead of investigating the exact dependency, we rather focus on comparing the performance of SCAL$^+$ to UCRL for different values of $\Gamma$. We consider

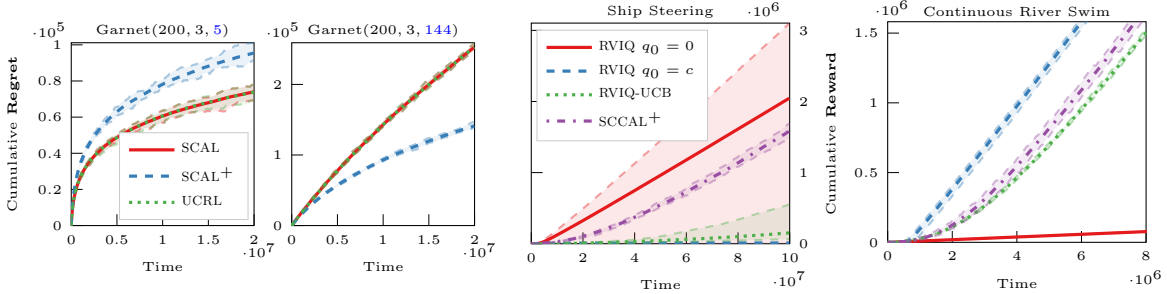

Figure 2: Cumulative regret (Garnet MDPs) and cumulative reward (continuous state MDPs). We report mean, max and min curves obtained over 50 independent runs.

the Garnet($S$, $A$, $\Gamma$) family [20] of random MDPs. In all the experiments we take $S = 200$, $A = 3$ and $c = 2$ and we guarantee the MDPs to be communicating by setting $p(s_0|s,a) \geq 0.01$ for every pair $(s,a)$ and an arbitrary state $s_0$. In order to provide a fair comparison of UCRL, SCAL and SCAL$^+$, we consider Hoeffdin-based confidence bounds with standardized constants: $\beta_k^r(s,a) = r_{\max}\sqrt{\mathcal{L}_k/N_k(s,a)}$ and $\beta_p(s,a) = \sqrt{\Gamma\mathcal{L}_k/N_k(s,a)}$ with $\mathcal{L}_k = \log(SA/\delta_k)/4$ for UCRL and SCAL, and $b_k(s,a) = \beta_k^r(s,a) + c\beta_k^p(s,a)$ for SCAL$^+$. Since Garnet($S$, $A$, $\Gamma$) defines a distribution over MDPs, we evaluate the algorithms on the MDP with median bias span (since the distribution shows relatively long tails, see App. E). According to the theoretical analysis, the per-episode regret of UCRL scales as $O(sp\,(h_k)\,\sqrt{\Gamma})$, where $sp\,(h_k)$ is the span of the optimistic MDP, while SCAL$^+$ has regret $O(c\sqrt{\Gamma})$, where $c$ is an upper-bound on $sp\,(h^*)$. While in the worst case $sp\,(h_k) \leq D$, in the MDP we selected, UCRL always generates optimistic MDPs with span $sp\,(h_k)$ smaller than $sp\,(h^*) \leq c$. In this favorable case for UCRL, the only hope for SCAL$^+$ to achieve better performance is if the tighter optimism translates into a per-episode regret of $O(c)$, with no dependency on $\Gamma$. This is indeed what we observed empirically. When $\Gamma = 5$, as expected, UCRL outperforms SCAL$^+$ as $sp\,(h_k)\,\sqrt{\Gamma} \leq c$ for most of the episodes. On the other hand, when $\Gamma = 144$, the tighter optimism of SCAL$^+$ allows a faster convergence to the optimal solution compared to UCRL as $sp\,(h_k)\,\sqrt{\Gamma} \geq c$. Although this result does not provide a definite answer on whether and how the regret of SCAL$^+$ scales with $\Gamma$, it hints to the fact that tighter optimism does indeed translate to better empirical performance w.r.t. confidence-based algorithms such as UCRL.

As SCCAL$^+$ is the first implementable model-based algorithm with regret guarantees in continuous MDPs, we compare it to model-free heuristic variants. We consider RVI Q-learning [17] with either $\epsilon$-greedy and UCB [9] exploration.[3] Since Q-learning is model-free, it does not perform planning and updates the policy at each time step (the action selection is greedy w.r.t. the current estimate). Even in this case we harmonize the bonus such that $b(s,a) = \beta^r(s,a) + c\beta^p(s,a) + (r_{\max} + c)\rho_L S^{-\alpha}$. We use the same uniform discretization of the state space for all the algorithms. We considering a continuous version of the *RiverSwim* [7] discretized into $S = 50$ states ($\rho_L = \alpha = 1$, $c = 30$, $\mathcal{S} \subseteq \mathbb{R}$) and the *ShipSteering* domain [21] with $S = |\mathcal{I}| = 512$ discrete states ($\rho_L = 5, \alpha = 1$, $c = 1.5$, $\mathcal{S} \subseteq \mathbb{R}^3$) (see the App. E for *MountainCar* [22]). In both cases, RVIQ shows an unstable behaviour. In the *RiverSwim* it outperforms the other approaches when optimistically initialized (i.e., $q_0 = c$) while the same configuration fails to learn in the *ShipSteering*. Moreover, RVIQ with $q_0 = 0$ shows the ability to learn in the *ShipSteering* but also high variance. This undesired behavior is typical of unstable algorithms (we observed linear regret in some run). RVIQ-UCB is able to learn in the *RiverSwim* but not in the *ShipSteering*. The only stable algorithm in both domains is SCCAL$^+$.

## 6 Conclusion

We derive the first regret analysis of exploration bonus for average reward with discrete and continuous state space by leveraging on an upper-bound to the range of the optimal bias function to properly scale the bonus (as done in other settings). It is an open question whether an exploration bonus approach is still possible when no prior knowledge on the span of the optimal bias function is available [see e.g., 11, 23]. Despite the $\sqrt{\Gamma}$ improvement in the definition of the exploration bonus (i.e., optimism) compared to confidence-set-based algorithms, the final regret still scales with $\Gamma$ leaving it as an open question whether such dependency can be actually removed in non-ergodic MDPs.

## Footnotes

[1]The algorithm is reported in its general form, which applies to both discrete and continuous state space.

[2]See https://arxiv.org/abs/1705.07041.

[3]Refer to App. E.1 for details about RVIQ and RVIQ-UCB. There is no known regret bound for model-free algorithms in average reward, we think this is an interesting line of research for future work.

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
