[Supplementary Material]

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

[4]Sample complexity is a more natural metric then regret in discounted problems.

[5]The original bound of Jaksch et al. [11] has $\sqrt{S}$ instead of $\sqrt{\Gamma}$ but $\sqrt{\Gamma}$ can be easily achieved by replacing Hoeffding inequality by empirical Bernstein's inequality for transition probabilities.

[6]Fruit et al. [1, Lem. 2] showed that there may not exist a deterministic optimal policy for problem 8.

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

**Structure of the appendix.** We start with a review of the literature and positioning of this paper (App A). We then review the problem of planning under bias span constraint (App B). We present the proofs for the continuous case first (App. C), and then for the discrete case (App. D) since the latter can be viewed as a special case of the former. That way, we only need to highlight the main differences in the discrete case. Because of the continuous nature of the state space and the aggregation of states in the continuous case, extra care needs to be taken while using concentration inequalities compared to standard regret proofs in RL (see App. C.1.3 for more details). In App. E we report all the details of the experiments. Finally, in App. F we recall/prove all the necessary results from probability theory that we use in the regret proofs.

For sake of clarity we state the main terms of the analysis in the following table.

| Discrete case | |
|---|---|
| $\overline{M}_k = (\mathcal{S}, \mathcal{A}, \overline{p}_k, \overline{r}_k)$ | Empirical (MLE) MDP |
| | $\overline{r}_k(s,a) = \sum_{t=1}^{t_k-1} \frac{r_t(s_t,a_t)\mathbb{1}\left((s_t,a_t)=(s,a)\right)}{N_k(s,a)}$ |
| | $\overline{p}_k(s'\|s,a) = \frac{N_k(s,a,s')}{N_k(s,a)}$ |
| $\widehat{M}_k = (\mathcal{S}, \mathcal{A}, \widehat{p}_k, \widehat{r}_k)$ | Perturbed MDP with bonus |
| | $\widehat{r}_k(s,a) = \overline{r}_k(s,a) + b_k(s,a)$ |
| | $\widehat{p}_k(s'\|s,a) = \frac{N_k(s,a)\overline{p}_k(s'\|s,a)}{N_k(s,a)+1} + \frac{\mathbb{1}(s'=\overline{s})}{N_k(s,a)+1}$ |
| | $b_k(s,a) := c \cdot \min\left\{\beta_k^{sa} + \frac{1}{N_k(s,a)+1}; 2\right\} + r_{\max} \cdot \min\left\{\beta_k^{sa}; 1\right\}$ |
| $\widehat{M}_k^+ = (\mathcal{S}, \mathcal{A}^+, \widehat{p}_k^+, \widehat{r}_k^+)$ | Augmentation of $\widehat{M}_k$ |
| | $\mathcal{A}^+ = \mathcal{A} \times \{1, 2\}$ |
| | $\widehat{r}_k^+(s, a_i) = \widehat{r}_k(s,a) \cdot \mathbb{1}(i=1)$ |
| | $\widehat{p}_k^+(s'\|s, a_i) = \widehat{p}_k(s'\|s,a)$ |

| Continuous case | |
|---|---|
| $\rho_L, \alpha$ | Hölder continuity parameters |
| $\overline{M}_k^{ag} = (\mathcal{I}, \mathcal{A}, \overline{r}_k^{ag}, \overline{p}_k^{ag})$ | Empirical (MLE) MDP (discrete state space) |
| | $I_1 := \left[0, \frac{1}{S}\right]$ and $I_k = \left]\frac{k-1}{S}, \frac{k}{S}\right]$ for $k = 2, \ldots, S$ |
| | $\mathcal{I} := \{I_1, \ldots, I_S\}$ $(\|\mathcal{I}\| = S)$ |
| | $\overline{r}_k^{ag}(I, a) := \frac{1}{N_k(I,a)} \sum_{t=1}^{t_k-1} r_t(s_t, a_t)\mathbb{1}(s_t \in I, a_t = a)$ |
| | $\overline{p}_k^{ag}(J\|I, a) := \frac{\sum_{s'\in J}\sum_{s\in I} N_k(s,a,s')}{\sum_{s\in I} N_k(s,a)}$ |
| $\widehat{M}_k^{ag} = (\mathcal{I}, \mathcal{A}, \widehat{r}_k^{ag}, \widehat{p}_k^{ag})$ | Perturbed MDP with bonus (discrete state space) |
| | $b_k(J, a) := c \cdot \min\left\{\beta_k^{Ja} + \rho_L S^{-\alpha} + \frac{1}{N_k(J,a)+1}; 2\right\} + r_{\max} \cdot$ $\min\left\{\beta_k^{Ja} + \rho_L S^{-\alpha}; 1\right\}, \qquad \forall J \in \mathcal{I}$ |
| $\widehat{M}_k^{ag+} = (\mathcal{I}, \mathcal{A}, \widehat{r}_k^{ag+}, \widehat{p}_k^{ag+})$ | Augmentation of $\widehat{M}_k^{ag}$ (discrete state space) |
| $\overline{M}_k = (\mathcal{S}, \mathcal{A}, \overline{r}_k, \overline{p}_k)$ | Empirical MDP (continuous state space) |

$$\overline{r}_k(s,a) := \overline{r}_k^{ag}(I(s),a)$$

$\overline{p}_k(s'|s,a)$ is the Radon-Nikodym derivative of the cumulative density function $F(s) = \sum_{s' \leq s} \frac{\sum_{x \in I(s)} N_k(x,a,s')}{N_k(I(s),a)}$

$\widehat{M}_k = (\mathcal{S}, \mathcal{A}, \widehat{r}_k, \widehat{p}_k)$     Perturbed MDP with bonus (continuous state space)

$$b_k(s,a) := b_k(I(s),a)$$

$$\widehat{r}_k(s,a) := \overline{r}_k(s,a) + b_k(s,a)$$

$$\widehat{p}_k(s'|s,a) := \frac{N_k(I(s),a)\overline{p}_k(s'|s,a)}{N_k(I(s),a)+1} + \frac{S \cdot \mathbb{1}(s' \in I(\overline{s}))}{N_k(I(s),a)+1}$$

$\widehat{M}_k^+ = (\mathcal{S}, \mathcal{A}, \widehat{r}_k^+, \widehat{p}_k^+)$     Augmentation of $\widehat{M}_k$ (continuous state space)

$\widetilde{M}_k = (\mathcal{S}, \mathcal{A}, \widetilde{r}_k, \widetilde{p}_k)$     Approximation of true MDP $M^*$ to be piecewise constant on $\mathcal{I}$ (continuous state space)

$$\widetilde{r}_k(s,a) := \frac{1}{N_k(I(s),a)} \sum_{x \in I(s)} N_k(x,a)r(x,a)$$

$$\widetilde{p}_k(s'|s,a) := \frac{1}{N_k(I(s),a)} \sum_{x \in I(s)} N_k(x,a)p(s'|x,a)$$

## A    Extended Introduction and Related Work

While learning in an unknown environment, a reinforcement learning (RL) agent must trade off the *exploration* needed to collect information about the dynamics and reward, and the *exploitation* of the experience gathered so far to gain reward. The performance of an online learning agent is measured in terms of its cumulative regret, which compares the rewards accumulated by the agent to the rewards obtained by an optimal policy. A popular strategy to trade off exploration and exploitation and minimize regret is the *optimism in the face of uncertainty* (OFU) principle.

Optimistic approaches have been widely studied in the context of stochastic multi-armed bandit (MAB) problems and RL. A popular technique to ensure optimism is to use an *exploration bonus*. OFU-based bandit algorithms maintain optimistic estimates of the expected reward for each action $a$ by adding a high-probability *confidence bound* $b(a)$ to the empirical average reward $\widehat{r}(a)$, thus obtaining the optimistic reward $\widehat{r}(a) + b(a)$. The action with highest optimistic estimate is then played [see e.g., 24]. The confidence bound plays the role of an *exploration bonus*: the higher $b(a)$, the more likely $a$ will be explored. For instance, the Upper-Confidence Bound (UCB) algorithm uses $b(a) = \widetilde{\Theta}\big(r_{\max}/\sqrt{N(a)}\big)$ where $N(a)$ is the number of times action $a$ has been played and $r_{\max}$ is a bound to the range of the rewards.

In RL, the agent aims at maximizing the cumulative expected reward (i.e., value function) rather than the immediate reward as in MAB. As a consequence, the exploration bonus should be designed to obtain an upper-bound on the value function. While standard concentration inequalities can be used to derive confidence bounds on "local" estimation errors on the reward function and dynamics, it is crucial to study how these errors compound when computing the value function. Exploiting the recursive structure of the optimal value function, it is possible to bound how errors are amplified through repeated applications of the optimal Bellman operator. Strehl and Littman [7] analysed the *infinite-horizon $\gamma$-discounted setting* and derived PAC guarantees on the sample complexity of the Model Based Interval Estimation with Exploration Bonus (MBIE-EB) algorithm.[4] MBIE-EB plays the optimal policy of the estimated MDP where in each state-action pair $(s,a)$, a bonus $b(s,a) = \widetilde{\Theta}\left(\frac{r_{\max}}{1-\gamma}\sqrt{\frac{1}{N(s,a)}}\right)$ is added to estimated reward $\widehat{r}(s,a)$. As the exploration bonus is designed to bound estimation errors on the value function, it scales by its range $\frac{r_{\max}}{1-\gamma}$.

The exploration bonus approach has also been successfully applied to *finite-horizon problems* [8, 9]. In this setting, the planning horizon $H$ is known to the learning agent and the range of the

value function is $r_{\max}H$. A natural choice for the bonus is then $b(s,a) = \widetilde{\Theta}\big(r_{\max}H/\sqrt{N(s,a)}\big)$. UCBVI_1 introduced by Azar et al. [8] uses such a bonus and achieves near-optimal regret guarantees $\widetilde{O}\big(H\sqrt{SAT}\big)$ where $S$ and $A$ are the number of states and the number of actions of the unknown MDP. Refined versions of UCBVI_1 achieve a tighter regret of $\widetilde{O}\big(\sqrt{HSAT}\big)$ [8, 25, 9].

Both the finite-horizon and infinite-horizon discounted setting assume an *intrinsic horizon* (respectively $H$ and $\frac{1}{1-\gamma}$) known to the learning agent. Unfortunately, in many common RL problems it is not clear how to define $H$ or $\frac{1}{1-\gamma}$ and it is often desirable to set them as big as possible (e.g., in episodic problems, the time to the goal is not known in advance and random in general). As $H$ tends to infinity the regret of UCBVI_1 becomes linear, while as $\gamma$ tends to 1 the sample complexity (of MBIE-EB, etc.) tends to infinity. In this paper, we focus on the more natural infinite-horizon undiscounted setting [10, Chap. 8], which generalizes the two previous settings to the case where $H \to \infty$ and $\gamma \to 1$ respectively. Several algorithms implementing the OFU principle in this setting have been proposed in the literature [e.g., 11, 2, 12, 1, 13], but none of these approaches exploits the idea of an exploration bonus. Instead, they all maintain a confidence set on the MDP (i.e., on the reward function and dynamics) and select the MDP with largest optimal average reward. Since the true MDP is contained in the set of plausible MDPs, the solution of this optimization problem is optimistic. UCRL [11] achieves a regret of order[5] $\widetilde{O}\big(r_{\max}D\sqrt{\Gamma SAT}\big)$ where $D$ and $\Gamma$ are respectively the diameter of the true MDP and the maximum number of states reachable from any state. Fruit et al. [1] introduced SCAL, achieving an improved bound $\widetilde{O}\big(\min\{r_{\max}D,c\}\sqrt{\Gamma SAT}\big)$ when a known upper bound $c$ on the range of the optimal value function (i.e., bias function) is known to the learning agent.

In this paper, we introduce and analyse SCAL⁺, the first algorithm that relies on an exploration bonus to efficiently balance exploration and exploitation in the infinite-horizon undiscounted setting. Similar to the exploration bonus used in finite-horizon and discount setting, which depends on the knowledge of $\gamma$ or $H$, we follow the approach of Bartlett and Tewari [14], Fruit et al. [1] and we assume an upper-bound $c$ on the range of the optimal bias is known. The exploration bonus used by SCAL⁺ is thus $b(s,a) = \widetilde{\Theta}\big(\max\{c,r_{\max}\}/\sqrt{N(s,a)}\big)$. We prove that this bonus ensures optimism using a novel technical argument. We no longer use an inclusion argument (i.e., the true MDP is contained in a set of plausible MDPs) but we reason directly at the level of the Bellman operator. We show that the optimistic Bellman operator defined by the empirical MDP with optimistic reward $\widehat{r}(s,a) + b(s,a)$ dominates the Bellman operator of the true MDP when *applied to the optimal bias function*. This is sufficient to prove that the solution of the optimistic MDP is indeed (gain-)optimistic. This proof technique has two main advantages w.r.t. the inclusion argument. First, it directly applies to slightly perturbed empirical MDPs, without re-deriving confidence sets. Second, as we study the optimistic Bellman operator applied only to the optimal bias function (rather than all possible vector in $\mathbb{R}^S$), we save a factor $\sqrt{\Gamma}$ in designing the exploration bonus, compared to the (implicit) bounds obtained by algorithms relying on confidence sets on the MDP. In practice, this may limit *over-exploration* and lead to better empirical performance. Furthermore, as SCAL⁺ only solves the estimated MDP with optimistic reward, it is computationally cheaper than UCRL-based algorithms, which require computing the optimal policy for an *extended* MDP with a continuous action space defined by the confidence set over MDPs.

To further illustrate the generality of the exploration bonus approach, we also present SCCAL⁺, an extension of SCAL⁺ to continuous state MDPs. As in [2, 16], we require the reward and transition functions to be Hölder continuous with parameters $\rho_L$ and $\alpha$. SCCAL⁺ is also the first implementable algorithm in continuous problem with theoretical guarantees (existing algorithms with theoretical guarantees such as UCCRL [2] cannot be implemented). The main result of the paper can be summarized as follows.

**Theorem 3** (informal). *For any MDP with $S$ states, $A$ actions and $\Gamma$ next states, the regret of SCAL⁺ is bounded with high probability by $\widetilde{O}\left(\max\{c,r_{\max}\}\sqrt{\Gamma SAT}\right)$. For any "smooth" MDP with smoothness parameters $\rho_L$ and $\alpha$, 1-dimensional state space $\mathcal{S} = [0,1]$ and $A$ actions, the regret of SCCAL⁺ is bounded with high probability by $\widetilde{O}\left(\max\{c,r_{\max}\}\rho_L\sqrt{A}T^{(\alpha+2)/(2\alpha+2)}\right)$.*

The regret bound of SCAL⁺ (resp. SCCAL⁺) matches the one of SCAL (resp. UCCRL). Surprisingly, the better optimism introduced by SCAL⁺ compared to SCAL and UCRL (i.e., the exploration bonus is smaller by a factor $\sqrt{\Gamma}$) is not reflected in the final regret bound with the current statistical analysis. We isolate and discuss where the term $\sqrt{\Gamma}$ appears in the proof sketch of Sect. 3.3. Azar et al. [8], Jin et al. [9] managed to remove the $\sqrt{\Gamma}$ term in the finite-horizon setting, while recently Ortner [15] derived an algorithm achieving $O\left(\sqrt{t_{\mathrm{mix}}SAT}\right)$ regret bound under the assumption that the true MDP is *ergodic* ($t_{\mathrm{mix}}$ denotes the maximum *mixing time* of any policy). It remains an open question if a regret scaling with $\sqrt{S}$ (instead of $\sqrt{\Gamma S}$) can be achieved in the infinite-horizon case without any ergodicity assumption (infinite mixing time for example). Adapting the proof techniques of the finite-horizon setting does not seem straightforward either as (among other things) the two definitions of regret do not match and differ by a linear term.

## B   Planning under span constraint

In this section we introduce and analyze the problem of planning under bias span constraint, i.e., maximizing the gain among policies $\pi$ satisfying $sp\left(h^\pi\right) \leq c$. This problem is at the core of the proposed algorithms (SCAL⁺ and SCCAL⁺) for exploration-exploitation. Formally:

$$g_c^*(M) := \sup_{\pi \in \Pi_c(M)} \{g^\pi\}, \tag{8}$$

where $M$ is any MDP (with discrete or continuous state space) s.t. $\Pi_c(M) := \{\pi \in \Pi^{\mathrm{SR}} : sp\left(h^\pi\right) \leq c \wedge sp\left(g^\pi\right) = 0\} \neq \emptyset$.[6] This problem is a slight variation of the bias-span constrained problem considered by [14, 2, 16], for which no known-solution is available. On the other hand, problem 8 has been widely analysed by Fruit et al. [1].

Problem 8 can be solved using ScOpt [1], a version of (relative) value iteration [10, 26], where the optimal Bellman operator is modified to return value functions with span bounded by $c$, and the stopping condition is tailored to return a constrained-greedy policy with near-optimal gain. Given $v \in \mathbb{R}^S$ and $c \geq 0$, we define the value operator $T_c : \mathbb{R}^S \to \mathbb{R}^S$ as

$$T_c v = \Gamma_c L v = \begin{cases} Lv(s) & \forall s \in \overline{\mathcal{S}}(c, v) \\ c + \min_s\{Lv(s)\} & \forall s \in \mathcal{S} \setminus \overline{\mathcal{S}}(c, v) \end{cases} \tag{9}$$

where $\overline{\mathcal{S}}(c, v) = \{s \in \mathcal{S} | Lv(s) \leq \min_s\{Lv(s)\} + c\}$ and $\Gamma_c$ is the span constrain projection ("truncation") operator (see [1, App. D] for details). In other words, operator $T_c$ applies a *span truncation* $\Gamma_c$ to the one-step application of $L$, which guarantees that $sp\left(T_c v\right) \leq c$. Given a vector $v_0 \in \mathbb{R}^S$ and a reference state $\overline{s}$ ScOpt implements relative value iteration where $L$ is replaced by $T_c$: $v_{n+1} = T_c v_n - T_c v_n(\overline{s})e$. We can now state the convergence guarantees of ScOpt [see 1, Lem. 8 and Thm. 10].

**Proposition 2.** *Given an MDP $M$ such that **1)** the associated optimal Bellman operator $L$ is a $\gamma$-span-contraction; **2)** all policies are unichain; **3)** the operator $T_c$ is globally feasible at any vector $v \in \mathbb{R}^S$ such that $sp\left(v\right) \leq c$, i.e., for all $s \in \mathcal{S}$, $\min_{a \in \mathcal{A}}\{r(s, a) + p(\cdot|s, a)^\mathsf{T} v\} \leq \min_{s'}\{Lv(s')\} + c$. Then:*

*(a)* Optimality equation: *there exists a solution $(g^+, h^+) \in \mathbb{R} \times \mathbb{R}^S$ to the optimality equation $T_c h^+ = h^+ + g^+ e$. Moreover, any solution $(g^+, h^+)$ satisfies $g^+ = g_c^*$.*

*(b)* Convergence: *for any initial vector $v_0 \in \mathbb{R}^S$, ScOpt converges to a solution $h^+$ of the optimality equation, and $\lim_{n \to +\infty} T_c^{n+1} v_0 - T_c^n v_0 = g^+ e$.*

We also recall that the operator $T_c$ satisfies the following property as a direct consequence of Fruit et al. [1, Lem. 8].

**Proposition 3** (Dominance). *If there exists $(g, h)$ satisfying $T_c h \geq h + ge$ then $g_c \geq g$.*

*Proof.* By induction, using the monotonicity and "linearity" of $T_c$ [1, Lemma 16 (a) & (c)], we have that $\forall n \in \mathbb{N}$, $\left(T_c\right)^{n+1} h \geq \left(T_c\right)^n h + ge$. By Prop. 2, $\lim_{n \to +\infty} \left(T_c\right)^{n+1} h - \left(T_c\right)^n h = g_c$. Taking the limit when $n$ tends to infinity in the previous inequality yields: $g_c \geq g$. □

We finally conclude this section by formally showing that the 3 assumptions stated earlier (Prop. 1 and 2) hold for the discrete MDP $\widehat{M}_k^+$ defined in Sec. 3.1.

**Proof of Prop. 1.** We follow the arguments of Fruit et al. [1, Sec. 6] for SCAL. We denote by $\widehat{L}^+$, $\widehat{L}$ and $L$ the optimal Bellman operators of $\widehat{M}_k^+$, $\widehat{M}_k$ and $M^*$ respectively. Similarly, we denote by $\widehat{T}_c^+$, $\widehat{T}_c$ and $T_c$ the "truncated" Bellman operators of these MDPs.

*1) Contraction of $\widehat{L}^+$.* The small bias in the definition of $\widehat{p}_k$ ensures that the "*attractive*" state $\overline{s}$ is reached with non-zero probability from any state-action pair $(s, a_i)$ implying that the *ergodic coefficient* of $\widehat{M}_k^+$ defined as $\gamma_k = 1 - \min_{\substack{s,u \in \mathcal{S}, \\ a,b \in \mathcal{A}^+}} \left\{ \sum_{j \in \mathcal{S}} \min \{\widehat{p}_k(j|s,a), \widehat{p}_k(j|u,b)\} \right\}$ is smaller than $1 - \min_{s,a} \left\{ \frac{N_k(s,a,\overline{s})+1}{N_k(s,a)+1} \right\} < 1$ and thus $\widehat{L}^+$ is $\gamma_k$-contractive [10, Thm. 6.6.6].

*2) $\widehat{M}_k^+$ is unichain.* By construction, the attractive state $\overline{s}$ necessarily belongs to all *recurrent classes* of all policies implying that $\widehat{M}_k^+$ is unichain (i.e., all policies are unichain).

*3) Global feasibility of $\widehat{T}_c^+$.* Let $v \in \mathbb{R}^S$ such that $sp(v) \leq c$ and let $(s^*, a_i^*) \in \mathcal{S} \times \mathcal{A}^+$ be such that $\widehat{r}_k^+(s^*, a_i^*) + \widehat{p}_k^+(\cdot|s^*, a_i^*)^\mathsf{T} v = \min_{s \in \mathcal{S}} \left\{ \max_{a \in \mathcal{A}^+} \{\widehat{r}_k^+(s,a) + \widehat{p}_k^+(\cdot|s,a)^\mathsf{T} v\} \right\}$. For all $(s, a_2) \in \mathcal{S} \times \mathcal{A}^+$ we have:

$$\widehat{p}_k^+(\cdot|s, a_2)^\mathsf{T} v - \widehat{p}_k^+(\cdot|s^*, a_i^*)^\mathsf{T} v \leq \max_{s \in \mathcal{S}}\{v(s)\} - \min_{s \in \mathcal{S}}\{v(s)\} = sp(v) \leq c$$

and $\widehat{r}_k^+(s, a_2) = 0 \leq \widehat{r}_k^+(s^*, a_i^*)$. Therefore, for all $s \in \mathcal{S}$, $\min_{a_j \in \mathcal{A}^+} \{\widehat{r}_k^+(s, a_j) + \widehat{p}_k^+(\cdot|s, a_j)^\mathsf{T} v\} \leq \min_{s'} \left\{ \widehat{L}^+ v(s') \right\} + c$, i.e., $\widehat{T}_c^+$ is globally feasible at $v$ [1, Lemma 5]. The application of Prop. 2 concludes the proof.

# C  Continuous state MDPs: the analysis of SCCAL⁺

In all this section we say that a function $v : s \in \mathcal{S} \mapsto \mathbb{R}$ is *piece-wise constant on $\mathcal{I}$* when $\forall J \in \mathcal{I}, \forall s, s' \in J$ we have $v(s) = v(s')$ and we denote by $v(J)$ the joint value.

## C.1  High probability bound using the exploration bonus (proof of Lem. 3)

To begin with, we introduce a slightly *tighter* exploration bonus than the one defined in (6). Despite being always smaller, it is of the same order of magnitude and has a slightly more complex expression. We decided to simplify it in the main body of the paper for the sake of clarity (this simplification does not change the final regret bound, but may impact the empirical performances).

$$b_k(J, a) := c \cdot \min \left\{ \beta_k^{Ja} + \rho_L S^{-\alpha} + \frac{1}{N_k(J,a)+1}; 2 \right\} + r_{\max} \cdot \min \left\{ \beta_k^{Ja} + \rho_L S^{-\alpha}; 1 \right\} \quad (10)$$

where $\beta_k^{sa}$ is defined in Eq. 3 (with $J$ in place of $s$ in this case). We also recall that the terms $\overline{r}_k, \widehat{r}_k$, $\overline{p}_k$ and $\widehat{p}_k$ are defined in Def. 1 in the main body. Lem. 3 is a direct consequence of the following (more general) result:

**Lemma 4.** *Consider the estimated continuous MDP $\widehat{M}_k$ defined in Def. 1. For any $v : \mathcal{S} \mapsto \mathbb{R}$, define the term*

$$\Delta L_k(s, a, v) = \left| \overline{r}_k(s, a) - r(s, a) + \int_{\mathcal{S}} (\widehat{p}_k(s'|s, a) - p(s'|s, a)) v(s') \mathrm{d}s' \right|$$

*and the event $F_b(v) := \{\forall k \geq 1, \forall (s, a) \in \mathcal{S} \times \mathcal{A}, b_k(s, a) \geq \Delta L_k(s, a, v)\}$. Then, for any $v$ such that $sp(v) \leq c$, $\mathbb{P}(F_b(v)) \geq 1 - \frac{\delta}{5}$.*

**Proof of Lem. 4** We introduce an intermediate MDP $\widetilde{M}_k := (\mathcal{S}, \mathcal{A}, \widetilde{r}_k, \widetilde{p}_k)$ (with continuous state-space) defined for all pairs $(s,a) \in \mathcal{S} \times \mathcal{A}$ by:

$$\widetilde{r}_k(s,a) := \frac{1}{N_k(I(s),a)} \sum_{x \in I(s)} N_k(x,a) r(x,a)$$

$$\widetilde{p}_k(s'|s,a) := \frac{1}{N_k(I(s),a)} \sum_{x \in I(s)} N_k(x,a) p(s'|x,a)$$

$\widetilde{M}_k$ can be interpreted as an approximation of $M$ where $s \mapsto \widetilde{r}_k(s,a)$ and $s \mapsto \widetilde{p}_k(s'|s,a)$ are constrained to be piece-wise constant on $\mathcal{I}$ in a way that depends on the states visited before episode $k$. We then decompose $\widehat{p}_k - p$ and $\overline{r}_k - r$ as

$$\widehat{p}_k - p = (\widehat{p}_k - \overline{p}_k) + (\overline{p}_k - \widetilde{p}_k) + (\widetilde{p}_k - p) \quad \text{and} \quad \overline{r}_k - r = (\overline{r}_k - \widetilde{r}_k) + (\widetilde{r}_k - r) \quad (11)$$

and bound all the terms separately in the next sections. To simplify the analysis, we "recenter" the function $v$ and define $w(s) := v(s) - (\max\{v(s)\} + \min\{v(s)\})/2$ so that for all $s \in \mathcal{S}$, $w(s) \in [-c/2, c/2]$.

### C.1.1 Bounding the difference between $\widetilde{r}_k/\widetilde{p}_k$ and $r/p$

To bound the differences $\widetilde{r}_k(s'|s,a) - r(s,a)$ and $\int (\widetilde{p}_k(s'|s,a) - p(s'|s,a)) w(s') ds'$ we simply use the smoothness assumption on the reward and transition model (see Asm. 2). For all $(s,a) \in \mathcal{S} \times \mathcal{A}$ (using the triangle inequality):

$$|\widetilde{r}_k(s,a) - r(s,a)| \leq \frac{1}{N_k(I(s),a)} \sum_{x \in I(s)} N_k(x,a) \underbrace{|r(x,a) - r(s,a)|}_{\leq r_{\max} L S^{-\alpha} \text{ since } x \in I(s)} \leq r_{\max} \rho_L S^{-\alpha} \quad (12)$$

For the transition probability we have that (using the triangle inequality):

$$\left| \int_{\mathcal{S}} (\widetilde{p}_k(s'|s,a) - p(s'|s,a)) w(s') ds' \right| \leq c \int_{\mathcal{S}} |\widetilde{p}_k(s'|s,a) - p(s'|s,a)| ds'$$

$$= \frac{c}{N_k(I(s),a)} \sum_{x \in I(s)} N_k(x,a) \int_{\mathcal{S}} \underbrace{|p(s'|x,a) - p(s'|s,a)|}_{\leq \rho_L S^{-\alpha} \text{ since } x \in I(s)} ds' \quad (13)$$

$$\leq c \rho_L S^{-\alpha}$$

Note that all these inequalities hold with probability 1.

### C.1.2 Bounding the difference between $\widehat{p}_k$ and $\overline{p}_k$

Using the triangle inequality and the fact that $\int_{\mathcal{S}} \mathbb{1}(s' \in I(\overline{s})) ds' = \int_{I(\overline{s})} 1 ds' = |I(\overline{s})| = 1/S$ we have that for any $(s,a) \in \mathcal{S} \times \mathcal{A}$:

$$\left| \int_{\mathcal{S}} (\widehat{p}_k(s'|s,a) - \overline{p}_k(s'|s,a)) w(s') ds' \right| \leq \int_{\mathcal{S}} |\widehat{p}_k(s'|s,a) - \overline{p}_k(s'|s,a)| \cdot |w(s')| ds'$$

$$= \left| \frac{N_k(I(s),a)}{N_k(I(s),a) + 1} - 1 \right| \left| \int_{\mathcal{S}} \overline{p}_k(s'|s,a) \underbrace{|w(s')|}_{\leq c/2} ds' \right|$$

$$+ S \int_{\mathcal{S}} \frac{|w(s')| \mathbb{1}(s' \in I(\overline{s}))}{N_k(I(s),a) + 1} ds'$$

$$\leq \frac{c}{N_k(I(s),a) + 1} \quad (14)$$

### C.1.3 Bounding the difference between $\widetilde{r}_k/\widetilde{p}_k$ and $\overline{r}_k/\overline{p}_k$

Let's consider a fixed pair $(s,a) \in \mathcal{S} \times \mathcal{A}$ and a fixed aggregated state $J \in \mathcal{I}$. Our goal is to bound the differences $\int_{\mathcal{S}} (\widetilde{p}_k(s'|s,a) - \overline{p}_k(s'|s,a)) w(s') ds'$, $\int_J \widetilde{p}_k(s'|s,a) - \overline{p}_k(s'|s,a) ds'$ and

$\widetilde{r}_k(s,a) - \overline{r}_k(s,a)$. Since $\widetilde{p}_k$ and $\widetilde{r}_k$ are in some sense the expected values of $\overline{p}_k$ and $\overline{r}_k$, we would like to use concentration inequalities. In the case of a finite state space $\mathcal{S}$, Jaksch et al. [11, UCRL ] and Fruit et al. [1, SCAL ] use concentration inequalities that apply to *independent* random variables (r.v.). We argue that a more careful analysis is needed here since the states lie in an *uncountable* set. Indeed, the implicit assumption made about the RL model for UCRL and SCAL is that for each state-action pair $(s,a)$, the rewards (respectively next states) are sampled from an infinite *stack* of independent and identically distributed (i.i.d.) rewards (respectively next states). More precisely, each time the agent visits $(s,a)$, it receives a reward from the top of the stack of rewards associated to $(s,a)$ and moves to the state on the top of the stack of next states associated to $(s,a)$. The two samples are then withdrawn from their respective stacks (meaning that they cannot be popped again). For more details about why this is a valid model refer to [27, Section 4.4]. In the case where $\mathcal{S}$ and $\mathcal{A}$ are discrete sets (finite or countable), it is possible to use any concentration inequality for i.i.d. r.v. and then take a union bound over all "stacks" $(s,a)$. When $\mathcal{S}$ is uncountable however, the same argument cannot be used (the probability of an uncountable union of events is not even always defined). Moreover, the terms $\widetilde{r}_k$ and $\widetilde{p}_k$ are obtained using sampled from different states $x \in I(s)$ instead of a single state $s$. To overcome these technical problems, we use a variant of Doob's optional skipping [e.g., 28, Sec. 5.3, Lem. 4] and concentration inequalities for martingales (Azuma and Freedman inequalities). For the sake of completeness, the argument that we use is formalized (with detailed proofs) in App. F (Thm. 4), but it is of course not new.

For any $t \geq 0$, the $\sigma$-algebra induced by the past history of state-action pairs and rewards up to time $t$ is denoted $\mathcal{F}_t := \sigma(s_1, a_1, r_1, \ldots, s_t, a_t)$ where by convention $\mathcal{F}_0 = \sigma(\emptyset)$ and $\mathcal{F}_\infty := \sigma(\cup_{t \geq 0} \mathcal{F}_t)$. Let $\mathbb{F}$ denote the filtration $(\mathcal{F}_t)_{t \geq 0}$. We define the following adapted sequences and stopping times:

### 1) Adapted sequences:
We consider the following stochastic processes adapted to $\mathbb{F}$: $(w(s_t))_{t \geq 0}$ and $(r_{t-1}(s_{t-1}, a_{t-1}))_{t \geq 0}$ (with the conventions $r_{-1}(s_{-1}, a_{-1}) = r_0(s_0, a_0) = r_\infty(s_\infty, a_\infty) = 0$ and $w(s_0) = w(s_\infty) = 0$). Theses processes are bounded as $|w(s_t)| \leq 2 \times \|w\|_\infty \leq c$ and $|r_{t-1}(s_{t-1}, a_{t-1})| \leq r_{\max}$ for all $t \geq 0$.

### 2) Stopping times:
We define $\tau := (\tau_l)_{l \geq 0}$ s.t. $\tau_0 := 0$ and $\tau_{l+1} := \inf\{t_k > t > \tau_l : s_t \in I(s), a_t = a\}$ (we omit the dependency in $(s,a)$ in the notation $\tau_l$). For all $l \geq 0$ and for all $t \geq 0$, $\tau_l := \{\tau_l = t\} \in \mathcal{F}_t$ and so $\tau_l$ is a stopping time w.r.t. $\mathbb{F}$ (see Def. 3 in App. F). By definition for any $l \geq 0$, $\tau_l < \tau_{l+1}$ a.s. (i.e., $\tau$ is strictly increasing, see Lem. 9). We denote $\mathcal{G}_l := \mathcal{F}_{\tau_{l+1}}$ the $\sigma$-algebra at stopping time $\tau_{l+1}$ (see Def. 4 in App. F).

All the assumptions of Thm. 4 are satisfied and we have that $\forall s, a \in \mathcal{S} \times \mathcal{A}$ (using the compact notation $N_k^{sa} := N_k(I(s), a)$ for the sake of readibility)

$$\mathbb{P}\left(\forall k \geq 1, \left|\sum_{l=1}^{N_k^{sa}} \left(r_{\tau_l}(s_{\tau_l}, a_{\tau_l}) - \mathbb{E}\left[r_{\tau_l}(s_{\tau_l}, a_{\tau_l}) \big| \mathcal{G}_{l-1}\right]\right)\right| \leq r_{\max}\sqrt{N_k^{sa} \ln\left(\frac{2N_k^{sa}}{\delta}\right)}\right) \geq 1 - \delta$$

$$\mathbb{P}\left(\forall k \geq 1, \left|\sum_{l=1}^{N_k^{sa}} \left(w(s_{\tau_l+1}) - \mathbb{E}\left[w(s_{\tau_l+1}) \big| \mathcal{G}_{l-1}\right]\right)\right| \leq c\sqrt{N_k^{sa} \ln\left(\frac{2N_k^{sa}}{\delta}\right)}\right) \geq 1 - \delta$$

We now need to relate the above sums to $\int_{\mathcal{S}} \left(\widetilde{p}_k(s'|s,a) - \overline{p}_k(s'|s,a)\right) w(s') \mathrm{d}s'$ and $\widetilde{r}_k(s,a) - \overline{r}_k(s,a)$. By defintion of $\tau$, we can rewrite $\overline{r}_k$ and $\overline{p}_k$ as follows:

$$\overline{r}_k(s,a) = \frac{1}{N_k(I(s),a)} \sum_{l=1}^{N_k(I(s),a)} r_{\tau_l}(s_{\tau_l}, a_{\tau_l})$$

$$\int_J \overline{p}_k(s'|s,a)\mathrm{d}s' = \frac{1}{N_k(I(s),a)} \sum_{l=1}^{N_k(I(s),a)} \mathbb{1}\left(s_{\tau_l+1} \in J\right)$$

It is also easy to verify that the following holds: $\mathbb{E}\left[w(s_{\tau_l+1}) \big| \mathcal{G}_{l-1}\right] = \int_{\mathcal{S}} p(s'|s_{\tau_l}, a_{\tau_l}) w(s') \mathrm{d}s'$, and $\mathbb{E}\left[r_{\tau_l}(s_{\tau_l}, a_{\tau_l}) \big| \mathcal{G}_{l-1}\right] = r(s_{\tau_l}, a_{\tau_l})$ (see Lem. 11 in App. F for a formal proof). As a result, we can

rewrite $\widetilde{r}_k$ and $\widetilde{p}_k$ as follows:

$$\widetilde{r}_k(s,a) = \frac{1}{N_k(I(s),a)} \sum_{x \in I(s)} N_k(x,a) r(x,a) = \frac{1}{N_k(I(s),a)} \sum_{l=1}^{N_k(I(s),a)} \mathbb{E}\big[r_{\tau_l}(s_{\tau_l}, a_{\tau_l})\big|\mathcal{G}_{l-1}\big]$$

$$\text{and} \int_{\mathcal{S}} \widetilde{p}_k(s'|s,a) w(s') \mathrm{d}s' = \frac{1}{N_k(I(s),a)} \sum_{x \in I(s)} N_k(x,a) \int_{\mathcal{S}} p(s'|x,a) w(s') \mathrm{d}s'$$

$$= \frac{1}{N_k(I(s),a)} \sum_{l=1}^{N_k(I(s),a)} \mathbb{E}\big[w(s_{\tau_l+1})\big|\mathcal{G}_{l-1}\big]$$

Then, we have that $\forall(s,a) \in \mathcal{S} \times \mathcal{A}$

$$\mathbb{P}\left(\forall k \geq 1, |\overline{r}_k(s,a) - \widetilde{r}_k(s,a)| \leq r_{\max}\beta_k^{sa}\right) \geq 1 - \frac{\delta}{10SA} \tag{15}$$

$$\mathbb{P}\left(\forall k \geq 1, \left|\int_{\mathcal{S}} \big(\overline{p}_k(s'|s,a) - \widetilde{p}_k(s'|s,a)\big) w(s') \mathrm{d}s'\right| \leq c\beta_k^{sa}\right) \geq 1 - \frac{\delta}{10SA} \tag{16}$$

To conclude, we take a union bound over all possible $(I(s),a) \in \mathcal{I} \times \mathcal{A}$ and $J \in \mathcal{I}$ (and over both inequalities 15 and 16). Note that we only need to take a union bound over $I(s) \in \mathcal{I}$ (and not $\mathcal{S}$) because $s \mapsto \widetilde{p}_k(\cdot|s,a)$ and $s \mapsto \widetilde{r}_k(s,a)$ are piecewise constant on $\mathcal{I}$ (and similarly for $\overline{p}_k$ and $\overline{r}_k$).

**Remark.** Note that for $N_k(I(s),a) = 0$ we have that $\beta_k^{sa} \geq 1$, thus the bound holds with probability one.

### C.1.4 Gathering all the terms

We first notice that $\int_{\mathcal{S}} \left(\widehat{p}_k(s'|s,a) - p(s'|s,a)\right) v(s') \mathrm{d}s' = \int_{\mathcal{S}} \left(\widehat{p}_k(s'|s,a) - p(s'|s,a)\right) w(s') \mathrm{d}s'$ since $w$ and $v$ are equal up to a constant shift and $\int_{\mathcal{S}} \widehat{p}_k(s'|s,a) \mathrm{d}s' = \int_{\mathcal{S}} p(s'|s,a) \mathrm{d}s' = 1$. Gathering equations (13), (14) and (16) we have that with probability at least $1 - \frac{\delta}{12}$, $\forall s, a, k$,

$$\left|\int_{\mathcal{S}} \left(\widehat{p}_k(s'|s,a) - p(s'|s,a)\right) v(s') \mathrm{d}s'\right| \leq c \cdot \min\left\{\beta_k^{sa} + \rho_L S^{-\alpha} + \frac{1}{N_k(I(s),a)+1}; 2\right\} \tag{17}$$

Gathering equations (15) and (12) we have that with probability at least $1 - \frac{\delta}{12}$

$$\forall s, a, k, \quad |\overline{r}_k(s,a) - r(s,a)| \leq r_{\max} \cdot \min\left\{\beta_k^{sa} + \rho_L S^{-\alpha}; 1\right\} \tag{18}$$

The lemma follows by applying the triangular inequality, a union bound (to have (18) and (17) to hold simultaneously) and by definition of the exploration bonus.

### C.2 Optimism (Proof of Lem. 2 and 3)

Let $\widehat{g}_k^{ag+} := g_c^*(\widehat{M}_k^{ag+})$ denote the solution of optimisation problem (4) on MDP $\widehat{M}_k^{ag+}$ (defined in Sec. 4). In this section we prove that:

**Lemma 5** (see Lem. 3). *Under event $F_b(h^*)$ (Lem. 4), for any episode $k \geq 1$, $\widehat{g}_k^{ag+} \geq g^*$.*

$\widehat{M}_k^{ag+}$ only has a finite number of states while the true MDP $M^*$ has an uncountable state-space. Thus, it is difficult to compare directly $\widehat{g}_k^{ag+}$ with $g^*$. To overcome this difficulty, we first compare $g^*$ with the gain of $\widehat{M}_k$ and then compare the latter to $\widehat{g}_k^{ag+}$.

**1. Optimism of $\widehat{M}_k$.** Let $\widehat{g}_k$ denote the solution of optimisation problem (4) on $\widehat{M}_k$. To prove that $\widehat{g}_k \geq g^*$ we can use Prop. 3 which –as explained in the main body of the paper– only requires to show that $\widehat{L}_k h^* \geq L h^*$ where $\widehat{L}_k$ is the optimal Bellman operator of $\widehat{M}_k$. Under event $F_b(h^*)$,

$$\forall s \in \mathcal{S}, \ \widehat{L}_k h^*(s) := \max_{a \in \mathcal{A}} \left\{\overline{r}_k(s,a) + b_k(s,a) + \int_{\mathcal{S}} \widehat{p}_k(s'|s,a) h^*(s') \mathrm{d}s'\right\}$$

$$\geq \max_{a \in \mathcal{A}} \left\{r(s,a) + \int_{\mathcal{S}} p(s'|s,a) h^*(s') \mathrm{d}s'\right\} = L h^*(s)$$

Therefore, $\widehat{g}_k \geq g^*$.

**2. Relationship between $\widehat{M}_k$ and $\widehat{M}_k^{ag+}$ (proof of Lem. 2).** We now show that $\widehat{g}_k^{ag+} = \widehat{g}_k$. Consider a piecewise-constant function $v_0$ on $\mathcal{S}$ (e.g., $v_0 = 0$) and a vector $u_0 \in \mathbb{R}^S$ satisfying $u_0(J) = v_0(J)$ for all $J \in \mathcal{I}$. We define the sequences $v_{n+1} := \widehat{T}_c v_n$ and $u_{n+1} := \widehat{T}_c^{ag+} u_n$. We show by induction that $u_n(J) = v_n(J)$ for all $n \geq 0$ and for all $J \in \mathcal{I}$. By definition it is true for $n = 0$ and for all $n \geq 0$:

$$
\int_{s \in \mathcal{S}} \widehat{p}_k(s'|s,a) v_n(s') \mathrm{d}s' = \sum_{J \in \mathcal{I}} \int_J \widehat{p}_k(s'|s,a) v_n(s') \mathrm{d}s'
$$
$$
= \sum_{J \in \mathcal{I}} v_n(J) \int_J \widehat{p}_k(s'|s,a) \mathrm{d}s' = \sum_{J \in \mathcal{I}} u_n(J)\, \widehat{p}_k^{ag}(J|I(s),a) \tag{19}
$$

where the last equality follows from (7) and the induction hypothesis. In addition $\widehat{r}_k(s,a)$ is also piecewise-constant on $\mathcal{I}$ and $\widehat{r}_k(s,a) = \widehat{r}_k^{ag}(I(s),a)$ for all $s \in \mathcal{S}$. Therefore, we have that $\widehat{L}_k^{ag} u_n(I(s)) = \widehat{L}_k v_n(s)$ for any $s \in \mathcal{S}$. Finally, the augmentation is not impacting the optimal Bellman operator (i.e., for any $v$, $\widehat{L}_k^{ag+} v = \widehat{L}_k^{ag} v$) so $\widehat{L}_k^{ag+} u_n(I(s)) = \widehat{L}_k v_n(s)$ and consequently $\widehat{T}_c^{ag+} u_n(I(s)) = \widehat{T}_c v_n(s)$ for any $s \in \mathcal{S}$. This shows that $v_{n+1}(J) = u_{n+1}(J)$ for all $J \in \mathcal{I}$ which concludes the proof by induction.

As shown by Fruit et al. [1, Theorem 10], $\lim_{n \to +\infty} v_{n+1}(J) - v_n(J) = \widehat{g}_k^{ag+}$ and $\lim_{n \to +\infty} u_{n+1}(J) - u_n(J) = \widehat{g}_k$ so that, under event $F_b$, $\widehat{g}_k^{ag+} = \widehat{g}_k \geq g^*$.

### C.3 Relaxation of the Exploration Bonus

In this section we introduce a new bonus $d_k(J,a)$ that will be used in the regret analysis:

$$
d_k(J,a) := c \cdot \min\left\{ \phi_k^{Ja} + \rho_L S^{-\alpha} + \frac{1}{N_k(J,a)+1}; 2 \right\} + r_{\max} \cdot \min\left\{ \beta_k^{Ja} + \rho_L S^{-\alpha}; 1 \right\} \tag{20}
$$

with $\beta_k^{Ja}$ defined in Eq. 3 (with $J$ in place of $s$) and

$$
\phi_k^{Ja} = 2\sqrt{\frac{S \ln\left(\frac{40 S^2 A N_k^+(J,a)}{\delta}\right)}{N_k^+(J,a)}} + \frac{4S \ln\left(\frac{40 S^2 A N_k^+(J,a)}{\delta}\right)}{N_k^+(J,a)} \geq \beta_{p,k}^{Ja} \tag{21}
$$

with $N_k^+(J,a) := \max\{1, N_k(J,a)\}$. Note that $d_k$ is a looser exploration bonus, i.e., $d_k(s,a) \geq b_k(J,a)$, for all $J,a,k$. However, it allows us to extend Lem. 4: instead of showing that for all $v$ such that $sp(v) \leq c$, $F_b(v)$ holds with high probability, we show that $\bigcap_{sp(v) \leq c} F_b(v)$ holds with high probability (when $b_k$ is replaced by $d_k$ and with the additional restriction that all the functions $v$ should be piece-wise constant).

**Lemma 6.** *Consider the estimated continuous MDP $\widehat{M}_k$ defined in Def. 1 and the term $\Delta L_k(s,a,v)$ defined in Lem. 4. Let's define the event*

$$
F_d := \{\forall v \text{ piece-wise constant s.t. } sp(v) \leq c, \ \forall k \geq 1, \forall (s,a) \in \mathcal{S} \times \mathcal{A}, \ d_k(s,a) \geq \Delta L_k(s,a,v)\}
$$

*Then, $\mathbb{P}(F_d) \geq 1 - \frac{\delta}{5}$.*

**Proof of Lem. 6** The proof follows the same steps as the proof of Lem. 4 and we do not repeat the derivation of inequality (18). Similarly to Eq. 13, for all $(s,a) \in \mathcal{S} \times \mathcal{A}$:

$$
\sum_{J \in \mathcal{I}} \left| \int_J (\widetilde{p}_k(s'|s,a) - p(s'|s,a)) \mathrm{d}s' \right| \leq \sum_{J \in \mathcal{I}} \int_J |\widetilde{p}_k(s'|s,a) - p(s'|s,a)| \mathrm{d}s'
$$
$$
= \frac{1}{N_k(I(s),a)} \sum_{x \in I(s)} N_k(x,a) \int_{\mathcal{S}} \underbrace{|p(s'|x,a) - p(s'|s,a)|}_{\leq \rho_L S^{-\alpha} \text{ since } x, s \in I(s)} \mathrm{d}s' \tag{22}
$$
$$
\leq \rho_L S^{-\alpha}
$$

and similarly to Eq. 14:

$$\sum_{J\in\mathcal{I}}\left|\int_J (\widehat{p}_k(s'|s,a) - \overline{p}_k(s'|s,a))\mathrm{d}s'\right| \leq \int_{\mathcal{S}}|\widehat{p}_k(s'|s,a) - \overline{p}_k(s'|s,a)|\,\mathrm{d}s' \leq \frac{1}{N_k(I(s),a)+1} \tag{23}$$

Let's consider the same sequence of stopping times $\tau$ as in App. C.1.3 and the additional adapted sequence $(\mathbb{1}(s_t \in J))_{t\geq 0}$ (with the convention $\mathbb{1}(s_0 \in J) = \mathbb{1}(s_\infty \in J) = 0$). This process is bounded as $|\mathbb{1}(s_t \in J)| \leq 1$. Using again Thm. 4, we can state that $\forall s,a,J \in \mathcal{S}\times\mathcal{A}\times\mathcal{I}$

$$\mathbb{P}\left(\forall k \geq 1, \left|\sum_{l=1}^{N_k^{sa}}\Big(\mathbb{1}(s_{\tau_l+1}\in J) - \mathbb{E}\big[\mathbb{1}(s_{\tau_l+1}\in J)\,\big|\mathcal{G}_{l-1}\big]\Big)\right|\right.$$

$$\left.\leq 2\sqrt{V_k(J)\ln\left(\frac{4N_k^{sa}}{\delta}\right)} + 4\ln\left(\frac{4N_k^{sa}}{\delta}\right)\right) \geq 1 - \delta \tag{24}$$

where $V_k(J) := \sum_{l=1}^{N_k^{sa}}\mathbb{V}\big(\mathbb{1}(s_{\tau_l+1}\in J)\big|\mathcal{G}_{l-1}\big)$. We now need to relate $\int_J \widetilde{p}_k(s'|s,a) - \overline{p}_k(s'|s,a)\mathrm{d}s'$ to the above sum and to provide an explicit formula for $V_k(J)$. By defintion of $\tau$, we can rewrite $\overline{p}_k$ as follows:

$$\int_J \overline{p}_k(s'|s,a)\mathrm{d}s' = \frac{1}{N_k(I(s),a)}\sum_{l=1}^{N_k(I(s),a)}\mathbb{1}(s_{\tau_l+1}\in J)$$

It is also easy to verify that $\mathbb{E}\big[\mathbb{1}(s_{\tau_l+1}\in J)\,\big|\mathcal{G}_{l-1}\big] = \int_J p(s'|s_{\tau_l},a_{\tau_l})\mathrm{d}s'$ (see Lem. 11 in App. F for a formal proof). As a result, we can rewrite $\widetilde{p}_k$ as follows:

$$\int_J \widetilde{p}_k(s'|s,a)\mathrm{d}s' = \frac{1}{N_k(I(s),a)}\sum_{l=1}^{N_k(I(s),a)}\mathbb{E}\big[\mathbb{1}(s_{\tau_l+1}\in J)\,\big|\mathcal{G}_{l-1}\big]$$

We can also give a more explicit expression for $V_k$:

$$\mathbb{V}\big(\mathbb{1}(s_{\tau_l+1}\in J)\big|\mathcal{G}_{l-1}\big) := \mathbb{E}\big[\underbrace{\mathbb{1}(s_{\tau_l+1}\in J)^2}_{=\mathbb{1}(s_{\tau_l+1}\in J)}\big|\mathcal{G}_{l-1}\big] - \mathbb{E}\big[\mathbb{1}(s_{\tau_l+1}\in J)\,\big|\mathcal{G}_{l-1}\big]^2$$

$$= \int_J p(s'|s_{\tau_l},a_{\tau_l})\mathrm{d}s' - \left(\int_J p(s'|s_{\tau_l},a_{\tau_l})\mathrm{d}s'\right)^2$$

implying:

$$V_k(J) = \sum_{l=1}^{N_k(I(s),a)}\underbrace{\left(1 - \int_J p(s'|s_{\tau_l},a_{\tau_l})\mathrm{d}s'\right)}_{\leq 1}\underbrace{\int_J p(s'|s_{\tau_l},a_{\tau_l})\mathrm{d}s'}_{\geq 0} \leq \sum_{x\in I(s)}N_k(x,a)\int_J p(s'|x,a)\mathrm{d}s'$$

Using Cauchy-Scwartz inequality (recall that $|\mathcal{I}| = S$)

$$\sum_{J\in\mathcal{I}}\sqrt{V_k(J)} \leq \sqrt{S\sum_{J\in\mathcal{I}}V_k(J)} \leq \sqrt{S\sum_{x\in I(s)}N_k(x,a)\sum_{J\in\mathcal{I}}\int_J p(s'|x,a)\mathrm{d}s'} = \sqrt{SN_k(I(s),a)}$$

Then, we have that $\forall(s,a)\in\mathcal{S}\times\mathcal{A}$ (the inequalities remain valid after replacing $N_k$ with $N_k^+$)

$$\mathbb{P}\left(\forall k\geq 1, \sum_{J\in\mathcal{I}}\left|\int_J \overline{p}_k(s'|s,a) - \widetilde{p}_k(s'|s,a)\mathrm{d}s'\right| \leq \phi_k^{I(s)a}\right) \geq 1 - \frac{\delta}{10SA} \tag{25}$$

where we took a union bound over and $J \in \mathcal{I}$. As in Sec. C.1.3, we can take a union bound over all possible $(I(s),a)\in\mathcal{I}\times\mathcal{A}$.

Let $v$ be a piecewise constant function on $\mathcal{I}$ s.t. $sp(v) \leq c$ and define $w(s) := v(s) - (\inf\{v(s)\} + \sup\{v(s)\})/2$. $w$ is also piecewise constant on $\mathcal{I}$ and for all $J\in\mathcal{I}$, $w(J) \in$

$[-c/2, c/2]$. Gathering equations (25), (22) and (23) we have with probability at least $1 - \frac{\delta}{12}$ that $\forall s, a, k$ and for all $v$ piece-wise constant on $\mathcal{I}$ such that $sp(v) \leq c$:

$$\sum_{J \in \mathcal{I}} \left| \int_J (\widehat{p}_k(s'|s,a) - p(s'|s,a)) v(s') \mathrm{d}s' \right| = \sum_{J \in \mathcal{I}} \left| w(J) \int_J (\widehat{p}_k(s'|s,a) - p(s'|s,a)) \mathrm{d}s' \right|$$

$$\leq \frac{c}{2} \sum_{J \in \mathcal{I}} \left| \int_J (\widehat{p}_k(s'|s,a) - p(s'|s,a)) \mathrm{d}s' \right|$$

$$\leq c \left( \phi_k^{I(s)a} + \rho_L S^{-\alpha} + \frac{1}{N_k(I(s),a)+1} \right) \quad (26)$$

Note also that $\sum_{J \in \mathcal{I}} \left| \int_J (\widehat{p}_k(s'|s,a) - p(s'|s,a)) \mathrm{d}s' \right| \leq 2$ so that another (trivial) bound is $2c$. The lemma follows by application of triangular inequality, union bound (to have (18) and (26) to hold simultaneously) and by definition of the exploration bonus $d_k(s,a)$.

## C.4 Regret Proof of SCCAL⁺ (Proof of Thm. 2)

In this section, we provide a complete proof for the regret bound of SCCAL⁺.

The arguments used in the proof of the following lemma will be used several times in the rest of the regret proof (we will not repeat these arguments for the sake of brevity). We define the episode at time $t$ as $k_t := \sup\{k \geq 1 : t \geq t_k\}$. Then,

**Lemma 7.** *With probability at least* $1 - \frac{\delta}{5}$

$$\forall T \geq 1, \sum_{t=1}^{T} r(s_t, a_t) \geq \sum_{t=1}^{T} \sum_{a \in \mathcal{A}} \pi_{k_t}(s_t, a) r(s_t, a) + 2r_{\max} \sqrt{T \ln \left( \frac{5T}{\delta} \right)}$$

*Proof.* For any $t \geq 0$, the $\sigma$-algebra induced by the past history of state-action pairs and rewards up to time $t$ is denoted $\mathcal{F}_t := \sigma(s_1, a_1, r_1, \ldots, s_t, a_t, s_{t+1})$ where by convention $\mathcal{F}_0 = \sigma(\emptyset)$ and $\mathcal{F}_\infty := \sigma(\cup_{t \geq 0} \mathcal{F}_t)$ (not that unlike in App. C.1.3, we include $s_{t+1}$ in the definition of $\mathcal{F}_t$). Let's consider the stochastic process $X_t = r_t(s_t, a_t) - \sum_{a \in \mathcal{A}} \pi_{k_t}(s_t, a) r(s_t, a)$. The term $\sum_{a \in \mathcal{A}} r(s_t, a) \pi_{k_t}(s_t, a)$ is $\mathcal{F}_{t-1}$-measurable and moreover

$$\mathbb{E}[r_t(s_t, a_t) | \mathcal{F}_{t-1}] = \sum_{a \in \mathcal{A}} \pi_{k_t}(s_t, a) r(s_t, a)$$

so that $\mathbb{E}[X_t | \mathcal{F}_{t-1}] = 0$. Since in addition $|X_t| \leq r_{\max}$, $(X_t, \mathcal{F}_t)_{t \geq 1}$ is a Martingale Difference Sequence (MDS) and we can apply Azuma's inequality (see for example Jaksch et al. [11, Lemma 10]):

$$\mathbb{P}\left( \sum_{t=1}^{T} r_t(s_t, a_t) \leq \sum_{t=1}^{T} \sum_{a \in \mathcal{A}} \pi_{k_t}(s_t, a) r(s_t, a) - r_{\max} \sqrt{4T \ln \left( \frac{5T}{\delta} \right)} \right) \leq \left( \frac{\delta}{5T} \right)^2 \leq \frac{\delta}{20T^2} \quad (27)$$

The lemma follows by taking a union bound over all possible values of $T \geq 1$ and noticing that $1 - \sum_{T=1}^{+\infty} \frac{\delta}{20T^2} = 1 - \frac{2\pi^2 \delta}{120} \geq 1 - \frac{\delta}{5}$. $\qquad \square$

Defining $\Delta_k = \sum_{s \in \mathcal{S}} \nu_k(s) \left( g^* - \sum_{a \in \mathcal{A}_{s_t}} r(s,a) \pi_k(s,a) \right)$ and using Lem. 7, it holds with probability at least $1 - \frac{\delta}{5}$ that: $\Delta(\mathrm{SCAL}^+, T) \leq \sum_{k=1}^{k_T} \Delta_k + 2r_{\max} \sqrt{T \ln \left( \frac{5T}{\delta} \right)}$. Note that $\nu_k(s)$ is the total number of observation of state $s$ in episode $k$ and is well-defined for $s$ lying in $[0,1]$. Finally, recall that for any subset $I \subseteq \mathcal{S}$, the sum $\sum_{s \in I} u_s$ is also well-defined as long as the collection $(u_s)_{s \in I}$ contains only a *finite* number of non-zero elements.

> ⚠ In this section we will abuse notation and write $p(\cdot|s,a)^\mathsf{T} v = \int_{\mathcal{S}} p(s'|s,a) v(s') \mathrm{d}s'$ for any probability density function $p$ defined on $\mathcal{S} = [0,1]$.

### C.4.1 Optimism and Bellman Equation

From now on we assume that $F_b(h^*) \cap F_d$ hold (see Lem. 4 and 6). Denote by $g_k$ and $h_k$ the gain and bias returned by SCOPT. By definition, $g_k := 1/2 \cdot (\max\{\widehat{T}_c^+ h_k - h_k\} + \min\{\widehat{T}_c^+ h_k - h_k\})$. The stopping condition of SCOPT [see 1] and Lem. 5 ensure that (since $F_b(h^*)$ holds)

$$g_k \geq g_c^*(\widehat{M}_k^{ag+}) - \frac{r_{\max}}{t_k} \overset{\text{Lem. 5}}{\geq} g^* - \frac{r_{\max}}{t_k}$$

implying

$$\Delta_k \leq r_{\max} \sum_{s \in \mathcal{S}} \frac{\nu_k(s)}{t_k} + \sum_{s \in \mathcal{S}} \nu_k(s) \underbrace{\left( g_k - \sum_{a \in \mathcal{A}_s} r(s,a)\pi_k(s,a) \right)}_{:=\Delta_k'(s)} \tag{28}$$

Note that we can associate a continuous piece-wise constant function $u_k : \mathcal{S} \mapsto \mathbb{R}$ to the discrete vector $h_k \in \mathbb{R}^S$: $u_k(s) := h_k(I(s))$, $\forall s \in \mathcal{S}$. Since event $F_d$ holds,

$$-r(s,a) \leq d_k(s,a) - \overline{r}_k(s,a) + (p(\cdot|s,a) - \widehat{p}_k(\cdot|s,a))^\mathsf{T} u_k.$$

Note that we cannot replace $d_k$ by $b_k$ (and invoke event $F_b(u_k)$ instead of $F_d$) since $u_k$ is correlated with $\widehat{p}_k$. By definition $\overline{r}_k(s,a) = \overline{r}_k^{ag}(I(s),a) = \widehat{r}_k(s,a) - b_k(s,a)$ and therefore,

$$\Delta_k'(s) \leq g_k - \sum_{a \in \mathcal{A}_s} \pi_k(s,a)\left( \underbrace{\widehat{r}_k(s,a)}_{:=\widehat{r}_k^{ag}(I(s),a)} + \widehat{p}_k(\cdot|s,a)^\mathsf{T} u_k \right)$$
$$+ \sum_{a \in \mathcal{A}_s} \pi_k(s,a)\left( \underbrace{b_k(s,a)}_{:=b_k(I(s),a)} + d_k(s,a) + p(\cdot|s,a)^\mathsf{T} u_k \right) \tag{29}$$

A direct consequence of the stopping condition used by SCOPT (see Thm. 18 of Fruit et al. [1]) is that: $\forall J \in \mathcal{I}$,

$$\left| g_k + h_k(J) - \sum_{a_i \in \mathcal{A} \times \{1,2\}} \pi_k(J,a_i) \left( \widehat{r}_k^{ag+}(J,a_i) - \widehat{p}_k^{ag+}(\cdot|J,a_i)^\mathsf{T} h_k \right) \right| \leq \frac{r_{\max}}{t_k} \tag{30}$$

Recall that by definition: $\pi_k(J,a) = \pi_k(J,a_1) + \pi_k(J,a_2)$, $\widehat{r}_k^{ag+}(J,a_i) \leq \widehat{r}_k^{ag}(J,a)$ (since we have $\widehat{r}_k^{ag+}(J,a_1) = \widehat{r}_k^{ag}(J,a)$ and $\widehat{r}_k^{ag+}(J,a_2) = 0$) and $\widehat{p}_k^{ag}(\cdot|J,a) = \widehat{p}_k^{ag+}(\cdot|J,a_i)$. We can thus write:

$$\sum_{a \in \mathcal{A}} \widehat{r}_k^{ag}(J,a)\pi_k(J,a) = \sum_{a \in \mathcal{A}} \sum_{i \in \{1,2\}} \widehat{r}_k^{ag}(J,a)\pi_k(J,a_i) \geq \sum_{a_i \in \mathcal{A} \times \{1,2\}} \widehat{r}_k^{ag+}(J,a_i)\pi_k(J,a_i)$$
$$\text{and } \sum_{a \in \mathcal{A}} \pi_k(s,a)\widehat{p}_k^{ag}(\cdot|J,a) = \sum_{a \in \mathcal{A} \times \{1,2\}} \pi_k(J,a_i)\widehat{p}_k^{ag+}(\cdot|J,a_i) \tag{31}$$

As in (19), we can easily show that $\widehat{p}_k(\cdot|s,a)^\mathsf{T} u_k = \widehat{p}_k^{ag}(\cdot|s,a)^\mathsf{T} h_k$. Plugging the two inequalities of (31) into (30), the fact that $u_k(s) = h_k(I(s))$, $\widehat{r}_k(s,a) = \widehat{r}_k^{ag}(I(s),a)$, and $\pi_k(s,a) = \pi_k(I(s),a)$, we obtain:

$$\forall s \in \mathcal{S}, \quad g_k - \sum_{a \in \mathcal{A}} \pi_k(s,a)\left( \widehat{r}_k(s,a) + \widehat{p}_k(\cdot|s,a)^\mathsf{T} u_k \right) \leq -u_k(s) + \frac{r_{\max}}{t_k} \tag{32}$$

Combining (32) with (29) we obtain

$$\Delta_k'(s) \leq \sum_{a \in \mathcal{A}_s} \pi_k(s,a)\left( d_k(s,a) + \underbrace{b_k(s,a)}_{\leq d_k(s,a)} + p(\cdot|s,a)^\mathsf{T} u_k \right) - u_k(s) + \frac{r_{\max}}{t_k} \tag{33}$$

Note that $d_k(s,a) \geq b_k(s,a)$ for any $(s,a) \in \mathcal{S} \times \mathcal{A}$ since the term $\phi_k^{Ia}$ (see Eq. 21) contains a $\sqrt{S}$ dependence that is not present in $\beta_k^{Ia}$. Since the dominant term is given by $d_k(s,a)$, we will consider

the following (loose) upper-bound $d_k(s,a) + b_k(s,a) \leq 2d_k(s,a)$ in the remaining of the proof. Gathering (28) and (33) we can now state that

$$\Delta_k \leq \underbrace{\sum_s \nu_k(s)\left(\left(\sum_a \pi_k(s,a)p(\cdot|s,a)^\mathsf{T} w_k\right) - w_k(s)\right)}_{:=\Delta_k^p}$$

$$+ 2\sum_{s,a} \nu_k(s)\pi_k(s,a)d_k(s,a) + 2r_{\max}\sum_{s\in\mathcal{S}} \frac{\nu_k(s)}{t_k} \tag{34}$$

where $w_k = u_k - (\min_s\{u_k(s)\} + \max_s\{u_k(s)\})/2$ is obtained by "recentering" $u_k$ around 0 so that $\|w_k\|_\infty = sp(w_k)/2 \leq c/2$. Then, similarly to what is done by Jaksch et al. [11, Sec. 4.3.2] and Fruit et al. [1, App. F.7, pg. 32], we have

$$\Delta_k^p = \sum_{t=t_k}^{t_{k+1}-1} \underbrace{\left(\sum_a \pi_k(s_t,a)\int p(s'|s_t,a)w_k(s')\mathrm{d}s'\right) - w_k(s_{t+1})}_{:=X_t} + \sum_{t=t_k}^{t_{k+1}-1} w_k(s_{t+1}) - w_k(s_t)$$

$$= \sum_{t=t_k}^{t_{k+1}-1} X_t + \underbrace{w_k(s_{t_k+1}) - w_k(s_{t_k})}_{\leq sp(w_k)\leq c}$$

Given the filtration $\mathcal{F}_t = \sigma(s_1,a_1,r_1,\ldots,s_{t+1})$, $X_t$ is an MDS since $|X_t| \leq c$ and $\mathbb{E}[X_t|\mathcal{F}_{t-1}] = 0$ since $\pi_{k_t}$ is $\mathcal{F}_{t-1}$-measurable. By using Azuma's inequality we have that with probability at least $1 - \frac{\delta}{5}$:

$$\forall T \geq 1, \quad \sum_{k=1}^{k_T} \Delta_k^p \leq 2c\sqrt{T\ln\left(\frac{5T}{\delta}\right)} + ck_T \tag{35}$$

with $k_T \leq SA\log_2\left(\frac{8T}{SA}\right)$ when $T \geq SA$ (see App. C.2 in [11]).

### C.4.2 Bounding the exploration bonus and summing up visit counts

Using again a martingale argument and Azuma's inequality (see Lem.7 and App. F.6 in [1]), since $d_k(s,a) \leq 2c + r_{\max} \leq 3\max\{c,r_{\max}\}$, we obtain with probability at least $1 - \frac{\delta}{5}$, that for all $T \geq 1$

$$\sum_{k=1}^m \sum_{s,a} \nu_k(s)\pi_k(s,a)d_k(s,a) \leq \sum_{k=1}^m \sum_{s,a} \nu_k(s,a)d_k(s,a) + 6\max\{c,r_{\max}\}\sqrt{T\ln\left(\frac{5T}{\delta}\right)} \tag{36}$$

We now gather inequalities (36), (35) into inequality (34) summed over all the episodes $k$ which yields that with probability at least $1 - \frac{4\delta}{5}$ (for $T \geq SA$):

$$\sum_{k=1}^m \Delta_k \leq 8\max\{c,r_{\max}\}\sqrt{T\ln\left(\frac{5T}{\delta}\right)} + cSA\log_2\left(\frac{8T}{SA}\right)$$

$$+ \underbrace{2r_{\max}\sum_{s\in\mathcal{S}} \frac{\nu_k(s)}{t_k}}_{=O(SA\ln(T))\ [1,\ \text{App. F.7}]} + 2\sum_{k=1}^m \sum_{s,a} \nu_k(s,a)d_k(s,a) \tag{37}$$

Let $\phi_k^{sa}$ as defined in Eq. 21, then

$$\sum_{k=1}^m \sum_{s,a} \nu_k(s,a)d_k(s,a) \leq \underbrace{\sum_{k=1}^m \sum_{s,a} \nu_k(s,a)r_{\max}\beta_k^{sa}}_{\text{see App. F.7 [1]}} + c\underbrace{\sum_{k=1}^m \sum_{s,a} \nu_k(s,a)\phi_k^{sa}}_{\text{see App. F.7 [1]}}$$

$$+ 2c\sum_{k=1}^m \sum_{s,a} \frac{\nu_k(s,a)}{N_k(s,a)+1} + \underbrace{(c+r_{\max})}_{\leq 2\max\{c,r_{\max}\}} \rho_L S^{-\alpha}T \tag{38}$$

We recall that [see e.g., 1, App. F.7]

$$r_{\max} \sum_{k=1}^{m} \sum_{s,a} \nu_k(s,a)\beta_k^{sa} = \widetilde{O}(r_{\max}\sqrt{SAT}) \quad \text{and} \quad c\sum_{k=1}^{m}\sum_{s,a} \nu_k(s,a)\phi_k^{sa} = \widetilde{O}(cS\sqrt{AT} + cS^2 A).$$

Similarly to what is done in [1, Eq. 58-60], we can write, $\forall T \geq 1$

$$\sum_{k=1}^{m}\sum_{s,a} \frac{\nu_k(s,a)}{N_k(s,a)+1} = O\left(SA\ln(T)\right) \tag{39}$$

### C.4.3 Completing the proof

Summing up all the contributions, we arrive at the conclusion that there exists a numerical constant $\chi$ such that with probability at least $1 - \delta$, for all $T \geq 1$, SCCAL$^+$ has a regret bounded by

$$\Delta(\text{SCCAL}^+, T) \leq \chi \cdot \left( \max\{r_{\max}, c\} \left( S\sqrt{AT\ln\left(\frac{T}{\delta}\right)} + S^2 A\ln^2\left(\frac{T}{\delta}\right) + \rho_L S^{-\alpha}T \right) \right)$$

We now set $S = \left(\alpha\rho_L\sqrt{\frac{T}{A}}\right)^{1/(\alpha+1)}$ so that

$$\Delta(\text{SCCAL}^+, T) = \widetilde{O}\bigg( \max\{r_{\max}, c\} \bigg( \underbrace{\max\left\{\alpha^{1/(\alpha+1)}, \alpha^{-\alpha/(1+\alpha)}\right\}}_{\leq 2,\ \forall\alpha\geq 0} \times$$

$$\times\ \rho_L^{1/(\alpha+1)} A^{\alpha/(2\alpha+2)}T^{(\alpha+2)/(2\alpha+2)} + \underbrace{\alpha^{2/(1+\alpha)}}_{\leq 2,\ \forall\alpha\geq 0}\rho_L^{2/(1+\alpha)} A^{\alpha/(1+\alpha)}T^{1/(1+\alpha)} \bigg)\bigg)$$

Finally, when $T \geq \rho_L^{2/\alpha} A$, the regret of SCCAL$^+$ is bounded by

$$\Delta(\text{SCCAL}^+, T) = \widetilde{O}\left(\max\{r_{\max}, c\}\rho_L^{1/(\alpha+1)} A^{\alpha/(2\alpha+2)}T^{(\alpha+2)/(2\alpha+2)}\right).$$

## D  Finite MDPs: the analysis of SCAL$^+$

In this section we analyse SCAL$^+$ by leveraging the results provided for the continuous state case. We define the bonus $b_k$ and state a lemma analogue to Lem. 4 and Lem. 6 combined (which among other things, implies that SCAL$^+$ is optimistic at each episode $k$). Finally, we provide the proof of the regret bound stated in Thm. 1.

### D.1  High probability bound using the exploration bonus (proof of Lem. 1)

To begin with, we introduce two variants of bonus that will be used for the regret proof (analogue to the bonuses used for continuous state spaces):

$$b_k(s,a) := c \cdot \min\left\{\beta_k^{sa} + \frac{1}{N_k(s,a)+1}; 2\right\} + r_{\max}\cdot\min\left\{\beta_k^{sa}; 1\right\}$$
$$d_k(s,a) := c \cdot \min\left\{\phi_k^{sa} + \frac{1}{N_k(s,a)+1}; 2\right\} + r_{\max}\cdot\min\left\{\beta_k^{sa}; 1\right\} \tag{40}$$

where $\beta_k^{sa}$ is defined as in Eq. 3 and

$$\phi_k^{sa} := 2\sqrt{\frac{(\Gamma(s,a)-1)\ln\left(\frac{40S^2 AN_k^+(s,a)}{\delta}\right)}{N_k^+(s,a)}} + \frac{4S}{N_k^+(s,a)}\ln\left(\frac{40S^2 AN_k^+(s,a)}{\delta}\right)$$

Notice that compared to $b_k$, $d_k$ explicitly depends on the number of states $S$ and next states $\Gamma$. Also, $d_k(s,a) \geq b_k(s,a)$ for any $(s,a) \in \mathcal{S} \times \mathcal{A}$. In the continuous case we could also have considered the number of next states $\Gamma(s,a)$ in the (true) aggregated MDP. However, this quantity is not very informative so we decided (for sake of clarity) to upper-bound it by the number of intervals $|\mathcal{I}| = S$.

**Lemma 8.** *Let $(g^*, h^*)$ be a solution of the optimality equation $Lh^* = h^* + g^*$ such that $sp(h^*) \leq c$. With probability at least $1 - \frac{\delta}{5}$, for all $T \geq 1$ and $k \geq 1$, for any $(s,a) \in \mathcal{S} \times \mathcal{A}$ and for any $v \in \mathbb{R}^S$ s.t. $sp(v) \leq c$ we have:*

$$(a) \ b_k(s,a) \geq \left| \overline{r}_k(s,a) - r(s,a) + (\widehat{p}_k(\cdot|s,a) - p(\cdot|s,a))^\mathsf{T} h^* \right|$$

$$(b) \ d_k(s,a) \geq \left| \overline{r}_k(s,a) - r(s,a) + (\widehat{p}_k(\cdot|s,a) - p(\cdot|s,a))^\mathsf{T} v \right|$$

*where $b_k$ and $d_k$ are defined as in Eq. 40. Events $(a)$ and $(b)$ hold individually with probability $1 - \frac{\delta}{5}$ and not simultaneously.*

*Proof.* We consider the discrete case as a special sub-case of the continuous one considered in Lem. 4. As explained in Sec. C.1.3, for the discrete case we can even use an independence argument based on the "stack of samples" idea [27, Sec. 4.4]. However, for sake the sake of brevity, we re-use the same MDS argument exploited in the continuous case. The main difference is that in the discrete case we do not need state aggregation and thus we replace every interval with a singleton function, i.e., $I(s) = s$, $\forall s \in \mathcal{S}$. Define $w := h^* - (\min\{h^*\} + \max\{h^*\})/2$ such that $w \in [-c/2, c/2]$. We decompose $\widehat{p}_k - p$ into $(\widehat{p}_k - \overline{p}_k) + (\overline{p}_k - p)$. As done in Eq. 14 (App. C.1.2), we can write that

$$\left| (\widehat{p}_k(\cdot|s,a) - \overline{p}_k(\cdot|s,a))^\mathsf{T} w \right| \leq \left| \frac{N_k(s,a)}{N_k(s,a) + 1} - 1 \right| \underbrace{\|\overline{p}_k(\cdot|s,a)\|_1}_{=1} \underbrace{\|w\|_\infty}_{\leq c/2} + \frac{|w(\overline{s})|}{N_k(s,a) + 1}$$

$$\leq \frac{c}{2} \left( 1 - \frac{N_k(s,a)}{N_k(s,a) + 1} + \frac{1}{N_k(s,a) + 1} \right) = \frac{c}{N_k(s,a) + 1} \tag{41}$$

In order to bound the term depending on $(\overline{p}_k - p)$ we use the same MDS argument as in App. C.1.3. You can consider $r$ equivalent to $\widetilde{r}$ defined in the continuous case since:

$$\widetilde{r}_k(s,a) = \frac{1}{N_k(\underbrace{I(s)}_{:=s}, a)} \sum_{x \in \underbrace{I(s)}_{:=s}} N_k(x,a) r(x,a) = r(s,a).$$

Similarly, we can prove that $\widetilde{p}_k(s'|s,a) = p(s'|s,a)$. Then, we consider the same adapted sequences, stopping times and predictable processes except from the fact that intervals are replaced by singletons (i.e., discrete states). As a consequence, (an analogue of) Lem. 11 holds. By following the same steps as in App. C.1.3, we can prove that with probability at least $1 - \frac{\delta}{10}$ (individually), for all $k \geq 1$ and $(s,a) \in \mathcal{S} \times \mathcal{A}$

$$\left| \overline{r}_k(s,a) - r_k(s,a) \right| \leq r_{\max} \sqrt{\frac{7 \ln\left( \frac{20 SAN_k^+(s,a)}{\delta} \right)}{N_k^+(s,a)}} := r_{\max} \beta_k^{sa}$$

$$\left| (\overline{p}_k(\cdot|s,a) - p(\cdot|s,a))^\mathsf{T} h^* \right| \leq c \sqrt{\frac{\ln\left( \frac{20 SAN_k^+(s,a)}{\delta} \right)}{N_k^+(s,a)}} := c \beta_k^{sa}$$

where we recall the $N_k^+(s,a) := \max\{1, N_k(s,a)\}$. We now consider the concentration of $(\widehat{p}_k - p)^\mathsf{T} v$ for which we need to use Freedman's inequality (see Thm. 4). Similarly to what done before, let $z = v - (\min\{v\} + \max\{v\})/2$ such that $(\widehat{p}_k - p)^\mathsf{T} v = (\widehat{p}_k - p)^\mathsf{T} z$. We start noticing that, Eq. 24 holds for the discrete case where we replace the adapted sequence $\mathbb{1}(s_{\tau_l+1} \in I)$ by $\mathbb{1}(s_{\tau_l+1} = s')$ and the conditional variance $V_k(J)$ by $V_k(s') = \sum_{l=1}^{N_k(s,a)} \mathbb{1}(\tau_l < t_k) \mathbb{V}(\mathbb{1}(s_{\tau_l+1} = s')|\mathcal{G}_{l-1})$. Furthermore, $\mathbb{V}(\mathbb{1}(s_{\tau_l+1} = s')|\mathcal{G}_{l-1}) = (1 - p(s'|s,a))p(s'|s,a)$ and

$$V_k(s') = \sum_{l=1}^{N_k(s,a)} (1 - p(s'|s_{\tau_l}, a_{\tau_l}))p(s'|s_{\tau_l}, a_{\tau_l}) \leq N_k(s,a)(1 - p(s'|s,a))p(s'|s,a)$$

As done in [1, App. F.7] we use Cauchy-Schwartz inequality to write that

$$\sum_{s' \in \mathcal{S}} \sqrt{p(s'|s,a)(1 - p(s'|s,a))} = \sum_{s' \in \mathcal{S}: \, p(s'|s,a) > 0} \sqrt{p(s'|s,a)(1 - p(s'|s,a))} \leq \sqrt{\Gamma(s,a) - 1}$$

where we recall that $\Gamma(s,a) := \|p(\cdot|s,a)\|_0$ is the support of $p$ in $(s,a)$. Then, as done in Lem. 6, we have that for any vector $z \in [-c/2, c/2]$, with probability at least $1 - \frac{\delta}{10}$ (after taking an union bound over $\mathcal{S} \times \mathcal{A}$) for all $(s,a) \in \mathcal{S} \times \mathcal{A}$ and $k \geq 1$

$$\left| \left( \overline{p}_k(\cdot|s,a) - p(\cdot|s,a) \right)^\mathsf{T} z \right| \leq \|z\|_\infty \sum_{s' \in \mathcal{S}} \left| \overline{p}_k(s'|s,a) - p(s'|s,a) \right|$$

$$\leq \frac{c}{2N_k^+(s,a)} \sum_{s' \in \mathcal{S}} \left( 2\sqrt{V_k(s') \ln\left(\frac{4N_k(s,a)}{\delta}\right)} + 4\ln\left(\frac{4N_k(s,a)}{\delta}\right) \right)$$

$$\leq c \left( \sqrt{\frac{(\Gamma(s,a)-1)\ln\left(\frac{48SAN_k^+(s,a)}{\delta}\right)}{N_k^+(s,a)}} + \frac{4S}{N_k^+(s,a)} \ln\left(\frac{48SAN_k^+(s,a)}{\delta}\right) \right)$$

We can also write with probability 1 that:

$$\left| \left( \widehat{p}_k(\cdot|s,a) - p(\cdot|s,a) \right)^\mathsf{T} w \right| \leq \widehat{p}_k(\cdot|s,a)^\mathsf{T} w + p(\cdot|s,a)^\mathsf{T} w \leq 2c$$

So we can take the minimum between the two upper-bounds. We also know that the difference in reward is bound by $r_{\max}$. □

In order to prove optimism we start noticing that the bonus $b_k(s,a)$ (see Lem. 8) implies that $\widehat{L}_k h^* \geq L h^*$. As a consequence, we can use Prop. 3 to show that $g_k^+ \geq g^*$ (same arguments as in with continuous state space).

### D.2 Regret Proof of SCAL⁺ (proof of Thm. 1).

The regret proof follows the same steps as for SCCAL⁺. The main difference resides in the fact that there is no need of state aggregation, thus simplifying the proof.

**Proof sketch.** In the following, all inequalities should be interpreted up to minor approximations and in high probability.Let $\nu_k(s,a)$ be the number of visits in $(s,a)$ during episode $k$ and $m$ be the total number of episodes. Using Lem. 1, we have:

$$\Delta(\text{SCAL}^+, T) \lesssim \sum_{k=1}^m \sum_{s,a} \nu_k(s,a) \left( g_k - \sum_a r(s,a)\pi_k(s,a) \right) \tag{42}$$

where $g_k$, $h_k$ and $\pi_k$ are respectively the gain, bias and policy returned by SCOPT [1]. SCOPT ensures that: $g_k + h_k(s) \simeq \sum_a \pi_k(s,a) \left( \widehat{r}_k(s,a) + \widehat{p}_k(\cdot|s,a)^\mathsf{T} h_k \right)$. By plugging this inequality into (42) we obtain two terms: $\overline{r}_k(s,a) - r(s,a) + b_k(s,a)$ and $(\widehat{p}_k(\cdot|s,a) - e_s)^\mathsf{T} h_k$. We can then introduce the true probability $(\widehat{p}_k(\cdot|s,a) - p(\cdot|s,a))^\mathsf{T} h_k + (p(\cdot|s,a) - e_s)^\mathsf{T} h_k$. Since $sp(h_k) \leq c$, the second term is of order $\widetilde{O}(c\sqrt{T} + cSA)$ when summed over $\mathcal{S}$, $\mathcal{A}$ and episodes $k$ [1, Eq. 56]. On the other hand, the term $(\widehat{p}_k(\cdot|s,a) - p(\cdot|s,a))^\mathsf{T} h_k$ is the dominant term of the regret and represents the error of using the estimated $\widehat{p}_k$ in place of $p$ in SCOPT. To bound this term we first compare the difference $(\widehat{p}_k - \overline{p}_k)^\mathsf{T} h_k$ which is not bigger than $c/(N_k(s,a) + 1)$. The remaining term is thus $(\overline{p}_k - p)^\mathsf{T} h_k$ (recall that $\overline{p}_k$ is the MLE of $p$). Since $h_k$ depends on $\overline{p}_k$, we cannot apply Hoeffding-Azuma inequality as done in the proof of Lem. 1. Instead we bound separately $\|\overline{p}_k(\cdot|s,a) - p(\cdot|s,a))\|_1 \lesssim \sqrt{\Gamma(s,a)} \beta_k^{sa}$ and $sp(h_k) \leq c$ which eventually introduce a $\sqrt{\Gamma}$ factor. It is worth pointing out that $\Gamma$ only appears due to statistical fluctuations that we cannot control, and not from the optimism (i.e., exploration bonus) that is explicitly encoded in the algorithm. For the reward we have $|\overline{r}(s,a) - r(s,a)| \leq r_{\max}\beta_k^{sa}$. As a consequence, we can approximately write that:

$$\Delta(\text{SCAL}^+, T) \lesssim \sum_{k=1}^m \sum_{s,a} \nu_k(s,a)\pi_k(s,a) \Big( \underbrace{b_k(s,a)}_{\leq d_k(s,a)} + \underbrace{(c\sqrt{\Gamma(s,a)} + r_{\max})\beta_k^{sa} + c/(N_k(s,a)+1)}_{:=d_k(s,a)} \Big)$$

The remaining terms can be bounded as in [1].

**Input:** Confidence $\delta \in ]0,1[$, $r_{\max}$, $\mathcal{I}$, $\mathcal{A}$, $c \geq 0$, $\rho_L$ and $\alpha$
**Initialization:** Observe intial state $s_0$ and set $\overline{s} = s_0$, $\overline{a} = a_0$, $q^0(s,a) = c$ for all $(s,a)$
**For** times $t = 1,2,...$ **do**
1. $a_t \in \arg\max_a\{q^t(s_t,a)\}$
2. Execute $a_t$, obtain reward $r_t$, and observe next state $s_{t+1}$ ($s_t = I(s_t)$, $s_{t+1} = I(s_{t+1})$).
3. $\alpha_t = \frac{c+1}{c+N_t(s_t,a_t)}$
4. Asynchronous update of the $Q$-function

$$q^{t+1}(s_t,a_t) = q^t(s_t,a_t) + \alpha_t \left( r_t + b_t(s_t,a_t) + \max_{a \in \mathcal{A}}\{q^t(s_{t+1},a)\} - q^t(s_t,a_t) - q^t(\overline{s},\overline{a}) \right)$$

$$q^{t+1}(s,a) = q^t(s,a), \qquad \forall(s,a) \neq (s_t,a_t)$$

5. $q^{t+1}(s_t,a_t) = \min\{c,q^{t+1}(s_t,a_t)\}$.
6. Increment counters $N_{t+1}(s,a) := N_t(s,a) + \mathbb{1}\left((s,a) = (s_t,a_t)\right)$ for all $s,a$.

Figure 3: RVIQ-UCB for continuous state (smooth) problems.

**Additional details.** By using the optimism of $\widehat{M}_k^+$, the stopping condition of SCOPT and the relationships between $\widehat{M}_k^+$ and $\widehat{M}_k$ (see Eq. 31), we can prove Eq. 34 for the discrete case. Note that the analysis of the cumulative contribution of the term $d_k(s,a)$ and $b_k(s,a)$ will lead to the following terms $\widetilde{O}(c\sqrt{\Gamma SAT})$ and $\widetilde{O}(c\sqrt{SAT})$, respectively. Since the dominant term is the one associated to $d_k$, even in this case we upper-bound $b_k$ by $d_k$.

From this point, we follow the same steps as in Sec. C.4. The only difference resides in Eq. 37 where the term $(c + r_{\max})\rho_L S^{-\alpha}T$ disappears since it depends on aggregation and/or smoothness. Finally, the regret bound in Thm. 1 follows by noticing that the order of the term $\sum_{k=1}^m \sum_{s,a} \nu_k(s,a)\phi_k^{sa}$ is $\widetilde{O}(\sqrt{\sum_{s,a}\Gamma(s,a)AT + S^2A})$.

As a consequence, there exists a numerical constant $\chi$ such that at least with probability $1 - \delta$ our algorithm SCAL$^+$ has a regret bounded by

$$\Delta(\text{SCAL}^+, T) \leq \chi \left( \max\{r_{\max},c\} \left( \sqrt{\left(\sum_{s,a}\Gamma(s,a)\right)T\ln\left(\frac{T}{\delta}\right)} + S^2A\ln^2\left(\frac{T}{\delta}\right) \right) \right)$$

# E  Experiments

We start describing the variants of Q-learning we considered in the experiments Then we report the configurations of algorithms and MDPs.

## E.1  RVI Q-learning

Relative Value Iteration is defined as
$$v^{n+1}(s) = \max_a \left\{ r(s,a) + p(\cdot|s,a)^\mathsf{T} v^n - v(\overline{s}) \right\}$$

where $\overline{s}$ is an arbitrary but fixed state. Under mild conditions, $v^n$ converges to the solution of the Bellman optimality equation $Lh = h + ge$ with $h(\overline{s}) = g$ (i.e., $v^n(\overline{s}) \to g$) [17]. This approach suggests a way of defining a *relative Q-factor iteration* algorithm which is defined as:

$$q^{n+1}(s,a) = r(s,a) - q^n(\overline{s},\overline{a}) + \sum_{s'} p(s'|s,a)\max_{a'}\{q^n(s',a')\}$$

with $(\overline{s},\overline{a})$ arbitrary but fixed.

The asynchronous RVI Q-learning algorithm [17] is a variant of Q-learning defined for average reward problems. At each iteration $n$ it computes an asynchronous update of the estimated Q-function of the current state action pair $(s_n,a_n)$ by using the observed transition and reward:

$$q^{n+1}(s,a) = q^n(s,a) + \alpha_n(s,a)\left(r_{n+1} + \max_{a \in \mathcal{A}}\{q^n(s_n',a)\} - q^n(s,a) - f(q^n)\right) \cdot \mathbb{1}\left((s,a) = (s_n,a_n)\right)$$

Figure 4: Distribution of the optimal bias span

Figure 5: Expected cumulative regret on 50 randomly generated MDPs. For each MDP we performed 5 runs for each algorithm.

where $s_{n+1} \sim p(\cdot|s_n, a_n)$ and $r_{n+1} \sim r(s_n, a_n)$ and $f$ is a Lipschitz function such that $f(e) = 1$ and $f(x + ce) = f(x) + c$ (see [17, Asm. 2.2]). In this work we consider $f(q) = q(\overline{s}, \overline{a})$. This algorithm is guaranteed to converge asymptotically under standard assumptions [17]. Note that $\max_a Q(s, a)$ is an estimate of the bias function $h(s)$ and $q(\overline{s}, \overline{a})$ an estimate of the gain.

RVIQ performs exploration by using an $\epsilon$-greedy strategy. In all the experiments we use

$$\epsilon_{t+1} = \frac{\epsilon_0}{\sqrt{N_t(s_t, a_t)}} \quad \text{and} \quad \alpha_t = \frac{1}{\sqrt{N_t(s_t, a_t)}}$$

where $\epsilon_0 = 20$ in order to force random exploration at the beginning of the experiment. Another parameter of RVIQ is the initial value used to fill the $q$-table. We consider both $q_0 = 0$ and $q_0 = c$. Q-learning with $q_0 = c$ is known as Optimistic Q-learning [29].

Finally, we use RVIQ to design an algorithm for exploration-exploitation inspired by the model-free approaches for finite-horizon problems [9]. We use an exploration bonus $b_t(s, a) = c \cdot \min\{\beta_t^{sa} + \rho_L S^{-\alpha}; 2\} + r_{\max} \cdot \min\{\beta_t^{sa} + \rho_L S^{-\alpha}; 1\}$. The algorithm is reported in 3. *We believe this is a reasonable algorithm but we want to stress that we do not have any theoretical guarantee for the regret of this algorithm.* The design of a model-free algorithm for efficient exploration-exploitation in average reward is an open problem.

Note that in the experiments we set $\overline{s}$ and $\overline{a}$ to the first observed state and performed action.

Figure 6: Cumulative regret in the Garnet domain with $\Gamma = 30$.

## E.2 Garnet

*Garnet* [20] is a family of randomly generated MDPs. The distribution over MDPs can be controlled by three parameters: number of states (i.e., $S = |\mathcal{S}|$), number of actions (i.e., $A = |\mathcal{A}|$) and branching factor (i.e., $\Gamma$). In the experiments we consider $S = 200$, $A = 3$ and $\Gamma \in \{5, 144\}$.

As we have seen in the main paper, the span of the optimal bias function plays an important role in the performance of the algorithms. As shown in Fig. 4, the distribution of the bias span has relatively long tails in the case of $\Gamma = 5$. In order to be more robust we evaluate the performance of the algorithms on the MDP with median bias span. The median bias span is $1.07$ and $1.59$ for $\Gamma = 144$ and $\Gamma = 5$, respectively. Her, we additionally report the performance of the algorithms when $\Gamma = 30$ (see Fig. 6). This confirms that SCAL$^+$ is able to exploit the tighter optimism and achieve a lower regret.

In the main paper we have reported the cumulative regret for the median MDP. Fig. 5 shows the expected cumulative regret w.r.t. the MDP distribution (in this case we use $c = 3.5$). As expected, the algorithms show the same behavior observed for the median MDP.

## E.3 Continuous RiverSwim

*RiverSwim* [7] is a classical domain for testing exploration algorithms for discrete problems. It is a stochastic chain having 6 states and 2 actions (left and right). The optimal policy is to always perform *right*. We preserve the idea of the domain but we generalize it to a continuous state space $\mathcal{S} = [0, 6]$. The domain is *not* episodic (the agent is never reset) and the initial position is $s_0 = 0$. At each time step the agent can move left or right by a factor $dx = 0.1$. The transitions are stochastics:

$$f(s, \text{left}) = \max\{0, s - dx - \sigma\epsilon\}$$

$$f(s, \text{right}) = \begin{cases} \max\{0, s - dx - \sigma\epsilon\} & \text{w.p. } 0.05 \\ s & \text{w.p. } 0.6 \\ \min\{6, s + dx + \sigma\epsilon\} & \text{w.p. } 0.35 \end{cases}$$

$$f(0, \text{right}) = \begin{cases} 0 & \text{w.p. } 0.4 \\ \max\{0, dx + \sigma\epsilon\} & \text{w.p. } 0.6 \end{cases}$$

$$f(6, \text{right}) = \begin{cases} \min\{6 - dx - \sigma\epsilon, 6\} & \text{w.p. } 0.4 \\ 6 & \text{w.p. } 0.6 \end{cases}$$

where $\epsilon \sim \mathcal{N}(0, 1)$ and $\sigma = dx/2$. The reward is zero everywhere except in $s = 0$ and $s = 6$ ($r(0, \text{left}) = 0.01$ and $r(6, \text{right}) = 1$).

In the *ContinuousRiverSwim* the state space is one-dimensional and we use 50 bins ($\rho_L = \alpha = 1$).

Figure 7: Expected cumulative reward in *MountainCar*.

## E.4 Ship Steering

The task is to steer a ship, which is cruising at constant speed, to a goal in minimum time [21]. The task is made not trivial by the presence of different water currents around the goal. The continuous state and action spaces are described by the 2-dimensional ship position and the scalar heading $(\mathcal{S} \subseteq \mathbb{R}^3)$. The following equations describes the continuous motion of the ship

$$\dot{x} = C(\cos(\phi) - y), \dot{y} = C\sin(\phi)$$

where $\phi$ is the ship heading. The start location is $(x_0, y_0) = (3.66, -1.86)$ and the goal region has a radius of $0.2$ km and the center is in $(0, 0)$. The speed is $C = 0.01$ km/h. The action is the steering angle (in degree) w.r.t. the current heading. The action set is $\mathcal{A} = \{-20, -10, -5, 0, 5, 10, 20\}$. Control decisions are made every 25s, at which time the ship changes heading instantaneously, but the change is affected by noise:

$$\phi_{t+1} = \phi_t + (a_t + \epsilon_t) \cdot \frac{2\pi}{360}$$

where $\epsilon_t$ is drawn uniformly from $[-2, 2]$. The state space is bounded as $\mathcal{S} \subseteq [0, 6] \times [-2, 2] \times [-\pi, \pi]$. The position of the ship is always clipped to this space. If the ship reaches position $x = 0$ but it is not in the goal (i.e., $y > 0.2$ or $y < -0.2$), the ship is reset to position $(5.5, 0)$.

In *ShipSteering* we use 8 bins for each state variable leading to 512 discrete states. Note that this discretization generates an MDP that is not communicating.

## E.5 Mountain Car

We consider the standard *MountainCar* [22]. The state space is two-dimensional and we use 29 bins for each dimension leading to 637 states after removing non-reachable states ($\rho_L = 5$ and $\alpha = 1$). We consider $c = 0.5$.

We report the results in Fig. 7 averaged over 50 runs. We report also max and min observed values. As mentioned in the main paper, RVIQ is unstable. Also in this case is able to learn with one type of initialization (here $q_0 = 0$) while shows high variance when initialized optimistically. SCCAL$^+$ is constantly the most stable algorithm.

# F Results of probability theory

## F.1 Reminder

We start by recalling some well-known properties of filtrations, stopping times and martingales [30, Chapter 2]. For simplicity, we use "a.s." to denote "almost surely" (i.e., with probability 1). In this section, we consider a probability space $(\Omega, \mathcal{F}, \mathbb{P})$. We call *filtration* any *increasing* (for the inclusion) sequence of sub-$\sigma$-algebras of $\mathcal{F}$ i.e., $(\mathcal{F}_n)_{n \in \mathbb{N}}$ where $\forall n \in \mathbb{N}, \mathcal{F}_n \subseteq \mathcal{F}_{n+1} \subseteq \mathcal{F}$. We

denote by $\mathcal{F}_\infty := \cup_{n \in \mathbb{N}} \mathcal{F}_n$. For any sub-$\sigma$-algebra $\mathcal{G} \subseteq \mathcal{F}$, we say that a real-valued random variable (r.v.) $X : \Omega \to \mathbb{R}^d$ is $\mathcal{G}$-measurable if for all borel sets $B \in \mathcal{B}(\mathbb{R}^d)$, $X^{-1}(B) \in \mathcal{G}$. We say that $X$ is $\mathcal{G}$-integrable if it is $\mathcal{G}$-measurable and $\mathbb{E}[|X|] < +\infty$ (componentwise). We call *stochastic process* any sequence of r.v. $(X_n)_{n \in \mathbb{N}}$. We say that the stochastic process $(X_n)_{n \in \mathbb{N}}$ is *adapted* to the filtration $(\mathcal{F}_n)_{n \in \mathbb{N}}$ if for all $n \in \mathbb{N}$, $X_n$ is $\mathcal{F}_n$-measurable. In this case, the sequence $(X_n, \mathcal{F}_n)_{n \in \mathbb{N}}$ is called an *adapted sequence*. If in addition, $X_n$ is integrable for all $n \in \mathbb{N}$ then we say that $(X_n, \mathcal{F}_n)_{n \in \mathbb{N}}$ is an integrable adapted sequence. We say that a stochastic process $(X_n)_{n \in \mathbb{N}}$ is almost surely:

1. *increasing* (resp. *strictly increasing*) if for all $n \geq N$, $\mathbb{P}(X_n \leq X_{n+1}) = 1$ (respectively $\mathbb{P}(X_n < X_{n+1}) = 1$),

2. *bounded* if there exists a universal constant $K$ such that for all $n \in \mathbb{N}$, $\mathbb{P}(X_n < K) = 1$,

**Definition 2** (Conditional expectation)**.** *Let $X$ be an $\mathcal{F}$-integrable r.v. with values in $\mathbb{R}^d$. Let $\mathcal{G} \subseteq \mathcal{F}$ be a sub-$\sigma$-algebra of $\mathcal{F}$. The* conditional expectation of $X$ given $\mathcal{G}$ *(denoted $\mathbb{E}[X|\mathcal{G}]$) is the (a.s. unique) r.v. that is $\mathcal{G}$-integrable and satisfies:*

$$\forall A \in \mathcal{G}, \quad \mathbb{E}[\mathbb{1}(A) \cdot \mathbb{E}[X|\mathcal{G}]] = E[\mathbb{1}(A) \cdot X]$$

**Proposition 4** (Law of total expextations)**.** *Let $X$ be an $\mathcal{F}$-integrable r.v. with values in $\mathbb{R}^d$. For any sub-$\sigma$-algebra $\mathcal{G} \subseteq \mathcal{F}$, $\mathbb{E}[\mathbb{E}[X|\mathcal{G}]] = \mathbb{E}[X]$.*

**Proposition 5.** *Let $X$ be an $\mathcal{F}$-integrable real-valued r.v. and $\mathcal{G} \subseteq \mathcal{F}$ a sub-$\sigma$-algebra. For any $\mathcal{G}$-integrable real-valued r.v. $Y$ s.t. $YX$ is also integrable we have $\mathbb{E}[YX|\mathcal{G}] = Y\mathbb{E}[X|\mathcal{G}]$.*

**Definition 3** (Stopping time)**.** *A random variable $\tau : \Omega \to \mathbb{N} \cup \{+\infty\}$ is* called stopping time *w.r.t. a filtration $(\mathcal{F}_n)_{n \in \mathbb{N}}$ if for all $n \in \mathbb{N}$, $\{\tau = n\} \in \mathcal{F}_n$.*

**Definition 4** ($\sigma$-algebra at stopping time)**.** *Let $\tau$ be a stopping time. An event prior to $\tau$ is any event $A \in \mathcal{F}_\infty$ s.t. $A \cap \{\tau = n\} \in \mathcal{F}_n$ for all $n \in \mathbb{N}$. The set of events prior to $\tau$ is a $\sigma$-algebra denoted $\mathcal{F}_\tau$ and called $\sigma$-algebra at time $\tau$:*

$$\mathcal{F}_\tau := \{A \in \mathcal{F}_\infty : \forall n \in \mathbb{N}, \ A \cap \{\tau = n\} \in \mathcal{F}_n\}$$

**Proposition 6.** *Let $\tau_1$ and $\tau_2$ be two stopping times w.r.t. the same filtration $(\mathcal{F}_n)_{n \in \mathbb{N}}$ s.t. $\tau_1 \leq \tau_2$ a.s. Then $\mathcal{F}_{\tau_1} \subseteq \mathcal{F}_{\tau_2}$.*

**Definition 5** (Stopped Process)**.** *Let $(X_n, \mathcal{F}_n)_{n \in \mathbb{N}}$ be an adapted sequence with values in $\mathbb{R}^d$. If $\tau$ is a stopping time w.r.t. the filtration $(\mathcal{F}_n)_{n \in \mathbb{N}}$, then the* process stopped at time $\tau$ *(denoted by $X_\tau$) is the r.v. defined as:*

$$\forall \omega \in \Omega, \quad X_\tau(\omega) := \sum_{n \in \mathbb{N}} X_n(\omega) \cdot \mathbb{1}(\tau(\omega) = n) \quad \text{(i.e., } X_\infty(\omega) = 0 \text{ by convention)}$$

**Proposition 7.** *$X_\tau$ –the process stopped at time $\tau$– is $\mathcal{F}_\tau$-measurable.*

**Definition 6** (Martingale difference sequence)**.** *An adapted sequence $(X_n, \mathcal{F}_n)_{n \in \mathbb{N}}$ is a* martingale difference sequence *(MDS for short) if for all $n \in \mathbb{N}$, $X_n$ is $\mathcal{F}_n$-integrable and $\mathbb{E}[X_{n+1}|\mathcal{F}_n] = 0$ a.s.*

**Proposition 8** (Azuma's inequality)**.** *Let $(X_n, \mathcal{F}_n)_{n \in \mathbb{N}}$ be an MDS such that $|X_n| \leq a$ a.s. for all $n \in \mathbb{N}$. Then for all $\delta \in ]0, 1[$,*

$$\mathbb{P}\left(\forall n \geq 1, \ \left|\sum_{i=1}^n X_i\right| \leq a\sqrt{n \ln\left(\frac{2n}{\delta}\right)}\right) \geq 1 - \delta$$

*Proof.* Azuma's inequality states that:

$$\mathbb{P}\left(\left|\sum_{i=1}^n X_i\right| \leq a\sqrt{\frac{n}{2} \ln\left(\frac{2}{\delta}\right)}\right) \geq 1 - \delta$$

We can then choose $\delta \leftarrow \frac{\delta}{2n^2}$ and take a union bound over all possible values of $n \geq 1$. The result follows by noting that $\sum_{n \geq 1} \frac{\delta}{2n^2} < \delta$. $\qquad \square$

**Proposition 9** (Freedman's inequality). *Let $(X_n, \mathcal{F}_n)_{n \in \mathbb{N}}$ be an MDS such that $|X_n| \leq a$ a.s. for all $n \in \mathbb{N}$. Then for all $\delta \in ]0, 1[$,*

$$\mathbb{P}\left(\forall n \geq 1, \; \left|\sum_{i=1}^{n} X_i\right| \leq 2\sqrt{\left(\sum_{i=1}^{n} \mathbb{V}\left(X_i \middle| \mathcal{F}_{i-1}\right)\right) \cdot \ln\left(\frac{4n}{\delta}\right) + 4a \ln\left(\frac{4n}{\delta}\right)}\right) \geq 1 - \delta$$

*Proof.* Freedman [31] showed that when $a = 1$:

$$\mathbb{P}\left(\forall n \geq 1, \; \sum_{i=1}^{n} X_i \geq \varepsilon, \; \sum_{i=1}^{n} \mathbb{V}\left(X_i \middle| \mathcal{F}_{i-1}\right) \leq k\right) \leq \exp\left(\frac{-\varepsilon^2}{2k + 2\varepsilon/3}\right)$$

Since $(-X_n, \mathcal{F}_n)_{n \in \mathbb{N}}$ is also an MDS, the above inequality holds also in absolute value (with a factor 2 appearing in front of the exponential term after taking a union bound). In order to reverse the inequality (i.e., replace $\varepsilon$ by $\delta$), we can use the same technique as Cesa-Bianchi and Gentile [32, Section 2]. Finally, to account for the case where $a \neq 1$ we can simply apply the result to $(X_n/a, \mathcal{F}_n)_{n \in \mathbb{N}}$. $\square$

### F.2 A useful concentration with optional skipping

In this section we prove a very simple theorem inspired by Doob's optional skipping [e.g., 28, Sec. 5.3, Lem. 4]. We start with useful definitions and lemmas.

**Lemma 9.** *Let $\tau_1$ and $\tau_2$ be two stopping times w.r.t. the same filtration $(\mathcal{F}_n)_{n \in \mathbb{N}}$. We say that $\tau_1 < \tau_2$ a.s. if $\mathbb{P}\left(\{\tau_1 < \tau_2\} \cup \{\tau_1 = \tau_2 = +\infty\}\right) = 1$. If $\tau_1 < \tau_2$ a.s. then $\mathcal{F}_{\tau_1 + 1} \subseteq \mathcal{F}_{\tau_2}$.*

*Proof.* If $\tau_1 < \tau_2$ then $\tau_1 + 1 \leq \tau_2$ since $\tau_1$ is an integer-valued r.v. If $\tau_1 = \tau_2 = +\infty$ then $\tau_1 + 1 = +\infty$ and so $\tau_1 + 1 = \tau_2$. In conclusion, $\tau_1 + 1 \leq \tau_2$ a.s. and so by Prop. 6, $\mathcal{F}_{\tau_1 + 1} \subseteq \mathcal{F}_{\tau_2}$. $\square$

**Definition 7.** *We say that a sequence of stopping times $(\tau_m)_{m \in \mathbb{N}}$ w.r.t. $(\mathcal{F}_n)_{n \in \mathbb{N}}$ is* strictly increasing *if $\tau_m < \tau_{m+1}$ a.s. for all $m \geq 0$.*

**Lemma 10.** *Let $(X_n, \mathcal{F}_n)_{n \in \mathbb{N}}$ be a bounded adapted sequence and let $(\tau_m)_{m \in \mathbb{N}}$ be a strictly increasing sequence of stopping times w.r.t. $(\mathcal{F}_n)_{n \in \mathbb{N}}$. For all $m \in \mathbb{N}$, define $Y_m := X_{\tau_m + 1} - \mathbb{E}\left[X_{\tau_m + 1} \middle| \mathcal{F}_{\tau_m}\right]$ and $\mathcal{G}_m := \mathcal{F}_{\tau_m + 1}$. Then, $(Y_m, \mathcal{G}_m)_{m \in \mathbb{N}}$ is an MDS.*

*Proof.* By assumption, for any $m \in \mathbb{N}$, $\tau_m < \tau_{m+1}$ a.s. and Prop. 6 implies that $\mathcal{F}_{\tau_m} \subseteq \mathcal{F}_{\tau_{m+1}}$. As a consequence, $(\mathcal{G}_m)_{m \in \mathbb{N}} = (\mathcal{F}_{\tau_m + 1})_{m \in \mathbb{N}}$ is a filtration. By Prop. 7 we know that $X_{\tau_m + 1}$ is $\mathcal{F}_{\tau_m + 1}$-measurable and Lem. 9 implies that $\mathcal{F}_{\tau_m + 1} \subseteq \mathcal{F}_{\tau_{m+1}} = \mathcal{G}_m$ so $X_{\tau_m + 1}$ is $\mathcal{G}_m$-measurable. Finally, $\mathbb{E}\left[X_{\tau_m + 1} \middle| \mathcal{F}_{\tau_m}\right]$ is $\mathcal{F}_{\tau_m}$-measurable by definition (see Def. 2). Therefore, $Y_m$ is $\mathcal{G}_m$-measurable.

Since by assumption $X_n$ is a.s. bounded ($\mathbb{P}\left(X_n < K\right) = 1$ for all $n \geq 0$), we can write a.s. (see Def. 5)

$$\left|X_{\tau_m + 1}\right| = \left|\sum_{n=0}^{+\infty} \mathbb{1}\left(\tau_m + 1 = n\right) \cdot X_n\right| \leq \sum_{n=0}^{+\infty} \mathbb{1}\left(\tau_m + 1 = n\right) \cdot |X_n| \leq K \sum_{n=0}^{+\infty} \mathbb{1}\left(\tau_m + 1 = n\right) = K$$

Thus, $X_{\tau_m + 1}$ is a.s. bounded hence integrable implying that $\mathbb{E}\left[X_{\tau_m + 1} \middle| \mathcal{F}_{\tau_m}\right]$ is well-defined (see Def. 2). Therefore, $Y_m$ is a.s. bounded and so integrable.

Finally, we can apply Prop. 5 and we obtain:

$$\mathbb{E}\left[Y_{m+1} \middle| \mathcal{G}_m\right] = \mathbb{E}\left[X_{\tau_m + 1} - \mathbb{E}\left[X_{\tau_m + 1} \middle| \mathcal{F}_{\tau_m}\right] \middle| \mathcal{F}_{\tau_m}\right] = \mathbb{E}\left[X_{\tau_m + 1} \middle| \mathcal{F}_{\tau_m}\right] - \mathbb{E}\left[X_{\tau_m + 1} \middle| \mathcal{F}_{\tau_m}\right] = 0$$

which concludes the proof. $\square$

**Theorem 4.** *Let $(X_n, \mathcal{F}_n)_{n \in \mathbb{N}}$ be an adapted sequence a.s. bounded by $a_1$ and let $(\tau_m)_{m \in \mathbb{N}}$ be a strictly increasing sequence of stopping times w.r.t. $(\mathcal{F}_n)_{n \in \mathbb{N}}$. If $(Y_m, \mathcal{G}_m)_{m \in \mathbb{N}}$ is defined as in*

*Lem. 10 then the following concentration inequalities hold:*

$$\mathbb{P}\left(\forall m \geq 1, \left|\sum_{i=1}^{m} Y_m\right| \leq a_1 \sqrt{m \ln\left(\frac{2m}{\delta}\right)}\right) \geq 1 - \delta$$

$$\mathbb{P}\left(\forall m \geq 1, \left|\sum_{i=1}^{m} Y_m\right| \leq 2\sqrt{\left(\sum_{i=1}^{m} \mathbb{V}\left(Y_i | \mathcal{G}_{i-1}\right)\right) \cdot \ln\left(\frac{4m}{\delta}\right) + 4a_1 \ln\left(\frac{4m}{\delta}\right)}\right) \geq 1 - \delta$$

*In particular for any sequence $(N_k)_{k \geq 1}$ of $\mathcal{F}$-measurable integer-valued r.v. $N_k : \Omega \to \mathbb{N} \setminus \{0\}$ the following inequality holds true*

$$\mathbb{P}\left(\forall k \geq 1, \left|\sum_{i=1}^{N_k} Y_m\right| \leq a_1 \sqrt{N_k \ln\left(\frac{2N_k}{\delta}\right)}\right) \geq 1 - \delta \quad \dots$$

*Proof.* The concentration inequalities follow from Lem. 10 and Azuma's and Freedman's inequalities. If the results hold for all $n \in \mathbb{N}$ and the r.v. $N_k$ takes values in $\mathbb{N}$, then the high probability event holds for all $N_k$ simultaneously. $\qquad \square$

### F.3 In the regret proof

For any $t \geq 0$, the $\sigma$-algebra induced by the past history of state-action pairs and rewards up to time $t$ is denoted $\mathcal{F}_t := \sigma\left(s_1, a_1, r_1, \dots, s_t, a_t\right)$ where by convention $\mathcal{F}_0 = \sigma\left(\emptyset\right)$ and $\mathcal{F}_\infty := \cup_{t \geq 0} \mathcal{F}_t$. Trivially, for all $t \geq 0$, $\mathcal{F}_t \subseteq \mathcal{F}_{t+1}$ and the filtration $(\mathcal{F}_t)_{t \geq 0}$ is denoted by $\mathbb{F}$. We recall that the sequence $(t_k)_{k \geq 1}$ (starting times of episodes $k \geq 1$) is formally defined by $t_1 := 1$ and for all $k \geq 1$,

$$t_{k+1} := 1 + \inf\left\{T \geq t > t_k : \sum_{u=t_k}^{t-1} \mathbb{1}(s_u \in I(s), a_u = a) \geq \sum_{u=0}^{t_k-1} \mathbb{1}(s_u \in I(s), a_u = a)\right\}.$$

where by convention $\inf\{\emptyset\} := T$. It is immediate to see that for all $t \geq 0$, $\{t_k = t\} \in \mathcal{F}_{t-1} \subseteq \mathcal{F}_t$ and so $t_k$ is a *stopping time* w.r.t. filtration $\mathbb{F}$ (see Def. 3).

The following lemma is used in App. C.1.3:

**Lemma 11.** *For all $l \geq 1$, we have:*

1. $\mathbb{E}\left[w^*(s_{\tau_l+1}) \big| \mathcal{G}_{l-1}\right] = \int_{\mathcal{S}} p(s' | s_{\tau_l}, a_{\tau_l}) w^*(s') \mathrm{d}s'$,

2. $\mathbb{E}\left[\mathbb{1}\left(s_{\tau_l+1} \in J\right) \big| \mathcal{G}_{l-1}\right] = \int_J p(s' | s_{\tau_l}, a_{\tau_l}) \mathrm{d}s'$,

3. *and* $\mathbb{E}\left[r_{\tau_l}(s_{\tau_l}, a_{\tau_l}) \big| \mathcal{G}_{l-1}\right] = r(s_{\tau_l}, a_{\tau_l})$.

*Proof.* To prove this result, we rely on the definition of conditional expectation (see Def. 2).
1) By Prop. 7, $(s_{\tau_l}, a_{\tau_l})$ is $\mathcal{G}_{l-1}$-measurable ($\mathcal{G}_{l-1} = \mathcal{F}_{\tau_l}$) and so $\int_{\mathcal{S}} p(s' | s_{\tau_l}, a_{\tau_l}) w^*(s') \mathrm{d}s'$ is $\mathcal{G}_{l-1}$-measurable too. Moreover, $\left|\int_{\mathcal{S}} p(s' | s_{\tau_l}, a_{\tau_l}) w^*(s') \mathrm{d}s'\right| \leq c/2$ a.s. so $\int_{\mathcal{S}} p(s' | s_{\tau_l}, a_{\tau_l}) w^*(s') \mathrm{d}s'$ is also integrable (and therefore $\mathcal{G}_{l-1}$-integrable).
2) We recall that for any stochastic process $(X_t)_{t \geq 0}$, we use the convention that $X_\infty = 0$ a.s. implying that $X_{\tau_l} = \sum_{t=0}^{+\infty} X_t \mathbb{1}\left(\tau_l = t\right)$ (see Def. 5). Usinng the law of total expectations (see Prop. 4) we have that $\forall A \in \mathcal{G}_{l-1}$,

$$\mathbb{E}\left[\mathbb{1}(A) \times w^*(s_{\tau_l+1})\right] = \sum_{t=0}^{+\infty} \mathbb{E}\left[\mathbb{1}(A \cap \{\tau_l = t\}) \times w^*(s_{\tau_l+1})\right]$$

$$= \sum_{t=0}^{+\infty} \mathbb{E}\left[\mathbb{E}\left[\mathbb{1}(\underbrace{A \cap \{\tau_l = t\}}_{\in \mathcal{F}_t}) \times w^*(\underbrace{s_{\tau_l+1}}_{=s_{t+1}}) \Big| \mathcal{F}_t\right]\right]$$

In the first equality, the fact that we can move the sum outside the expectation is a direct consequence of the dominated convergence theorem (for series) since

$$\sum_{t=0}^{+\infty} \mathbb{E}\big[\mathbb{1}(A \cap \{\tau_l = t\}) \times |w^*(s_{\tau_l+1})|\big] \leq c/2 \sum_{t=0}^{+\infty} \mathbb{E}\big[\mathbb{1}(A \cap \{\tau_l = t\})\big]$$

$$= c/2 \sum_{t=0}^{+\infty} \mathbb{P}\left(A \cap \{\tau_l = t\}\right) = c/2 \cdot \mathbb{P}(A) < +\infty$$

Under event $\{\tau_l = t\}$ we have that $s_{\tau_l+1} = s_{t+1}$ a.s. Moreover, $A \cap \{\tau_l = t\} \in \mathcal{F}_t$ since $\tau_l$ is a stopping time (see Def. 4) so by Prop. 5 we can move it outside the conditional expectation and we get:

$$\mathbb{E}\big[\mathbb{1}(A) \times w^*(s_{\tau_l+1})\big] = \sum_{t=0}^{+\infty} \mathbb{E}\Bigg[\mathbb{1}(A \cap \{\tau_l = t\}) \times \underbrace{\mathbb{E}\Big[w^*(s_{t+1})\Big|\mathcal{F}_t\Big]}_{=\int_{\mathcal{S}} p(s'|s_t,a_t)w^*(s')\mathrm{d}s'}\Bigg]$$

$$= \mathbb{E}\Bigg[\mathbb{1}(A) \times \underbrace{\sum_{t=0}^{+\infty} \mathbb{1}(\tau_l = t) \int_{\mathcal{S}} p(s'|s_t, a_t)w^*(s')\mathrm{d}s'}_{=\int_{\mathcal{S}} p(s'|s_{\tau_l},a_{\tau_l})w^*(s')\mathrm{d}s' \text{ (see Def. 5)}}\Bigg]$$

$$= \mathbb{E}\Bigg[\mathbb{1}(A) \times \int_{\mathcal{S}} p(s'|s_{\tau_l}, a_{\tau_l})w^*(s')\mathrm{d}s'\Bigg]$$

This proves the first inequality (see Def. 2). The second and third equality can be proved using the same technique. □