[Reviews · NeurIPS 2019]

Reviewer 1



Originality: learning weakly-communicating MDPs was already attempted in Bartlett and Tewari who give a very similar result to the one given in this paper (based on span-regularization rather than exploration bonus). Quality: The results seem sound. Clarity: The paper is well-written. Significance: The idea of exploration bonus has received a lot of attention in recent results. However, given the work of Bartlett and Tewari, the benefit of using exploration bonuses is mostly computational. --- Edit --- Having read the authors response, I agree that this is the first tractable algorithm for this setting. I have raised my score.

Reviewer 2



Originality: The work builds on the SCAL algorithm, but contains some novel aspects. In particular, the technique used to prove optimism (i.e. that the gain used to compute a policy is an upper bound on the true optimal gain) is different from UCRL and SCAL. While the latter solve an expanded version of the MDP, this paper shows that the value update operator T_c^+ (similar to the Bellman operator, but additionally projecting to a span constraint) is optimistic wrt its application to the optimal bias function. This is tighter by a \sqrt{\Gamma} factor, though the \sqrt{\Gamma} factor remains in the regret bound. The authors comment that it is an open question whether the \sqrt{\Gamma} dependency can be removed from the regret. This term is not present in the Politex algorithm (http://proceedings.mlr.press/v97/lazic19a/lazic19a.pdf) in tabular MDPs, though the dependence on T is suboptimal there. The analysis of SCCAL+ relies on similar assumptions and discretization as UCCRL [2], but has a practical implementation. Quality: The technical content of the paper seems correct, though I did not read the supplementary material. The experiments are small-scale, and SCAL+ is not always better than SCAL or UCRL. However it does perform best in a setting with large Gamma, which is to be expected based on the theory. Clarity: The paper is well-written, and the proof sketches and explanations make sense. It would be useful to explain the continuous state contributions better, including placing them in the context of previous work, and explaining why UCCRL cannot be implemented. Significance: While there is no improvement in regret results, the proposed algorithms seem practical and the new technique for showing optimism may be useful in future works. After reading the rebuttal, I am still in favor of acceptance, and my score remains the same.

Reviewer 3



This paper extends the SCAL algorithms for regret minimization in MDPs from the discounted and finite horizon cases to the average cost case. The algorithm has the "optimism in the face of uncertainty" flavor, and the authors establish a sqrt(\Gamma S A T) regret bound, matching the one for SCAL. While there is still a sqrt(gamma) gap to the lower bound, the improvement is appreciated, as is development of non confidence-region style algorithms. At a high level, the authors show how to add an optimism term to a simple empirical estimate of the MDP dynamics and control the regret. As a consequence, the algorithm is implementable if one has an approximate planning algorithm. Crucially, they need to assume an upper bound on the bias function. The algorithm is extended to continuous state spaces with a well known discretization, but the proofs are quite technical. The authors claim the first implementable algorithm for this setting with a regret bound. Finally, fairly comprehensive experiments are provided (and expanded on in the appendix), showing that the proposed algorithm is sane, though I wish the authors would provide some quantization about the stability of the algorithm on the two continuous settings. The presentation is OK. Most of the definitions are provided, and the notation is (maybe understandably) very intricate. The rational for the exploration bonus terms (3) is presented well in the paragraph starting 153, and I wish the authors would emphasize these points more. My biggest criticism is that the authors were sloppy with their definitions and left quite a few terms undefined. A few terms that I don't think the general NeurIPS community will be familiar with (and should probably be defined) include: Cesaro limit, ScOpt (provide a citation), unichain, etc. Overall, I think the technical contribution is strong. It advances an important and current problem (learning in MDPs) into the average cost setting, which has generally thought to be the most difficult. The algorithms studied, while extensively applied in the bandit literature, are not used as often in RL: it is good to see their application into a new domain. other things: 282: forgot to finish your sentence. I wist the experiment results were given more interpretation.

[Author Response · NeurIPS 2019]

We would like to thank the reviewers for their insightful comments.

**Reviewer 1.** We use the same assumption of knowing an upper bound to the optimal bias span as in (Bartlett and Tewari), but the algorithmic approach is very different. First, Fruit et al. ICML 2018 showed that the planning problem solved by REGAL.C is likely to be ill-posed and it is not clear how to derive a tractable algorithm to solve it (such an algorithm would likely be computationally inefficient). Furthermore, unlike (Fruit et al. ICML 2018), we introduce an exploration bonus formulation to the problem. This different algorithmic approach (the regret guarantees are indeed very similar) allows us to easily extend the algorithm to continuous MDPs and to achieve a "tighter" optimism, which results in a better empirical performance when $\Gamma$ is big enough.

**Reviewer 2.** The best known minimax regret lower bound for discrete communicating MDPs is $\sqrt{DSAT}$ and the proof relies on the construction of a family of adversarial MDPs for which $\Gamma = 2$ is fixed (unlike $D$, $S$ and $A$). It is still an open question whether a higher lower-bound can be shown (using a different family of adversarial MDPs for example) e.g., a lower-bound of order $\sqrt{DS\Gamma AT}$. If such a lower bound can be proved, then it is impossible to remove the dependence on $\Gamma$ in the upper-bound (but maybe such a lower-bound does not exist, this is the reason why we mention that it is not clear if it is possible to remove $\Gamma$ in the upper-bound or not). The comparison with the Politex paper is not simple due to the suboptimal dependence in T. The absence of explicit dependence in $\Gamma$ is due to the assumption of uniformly fast mixing MDP. As shown in Thm 5 there is a term of order $T^{1/2}k$ where $k$ is the mixing coefficient. This term hides the structural properties of the MDP. Moreover, the comparison is even more difficult due to the very suboptimal dependence on $T$ (ie $cT^{3/4} + \epsilon T$). The main contribution in the continuous case is a practical algorithm with theoretical guarantees. UCCRL cannot be implemented as it requires solving the same planning problem as in REGAL.C (optimistic span-regularized optimization). As argued in (Fruit et al. ICML 2018) this problem may be ill-posed and there is currently no known algorithmic solution. The exploration bonus approach makes it easy to manage the continuous case and its analysis follows by combining the discrete case with ideas from UCCRL original analysis.

**Reviewer 4.** We show empirically that SCCAL+ is a very stable algorithm: in the experiments we did not optimize any parameter, we simply harmonized the confidence intervals. In fact, the low variance in the performance obtained across different runs shows that the performance of SCCAL+ is very consistent (i.e., high initialization is sometimes much better than low and viceversa, depending on the problem). On the other hand, RVI Q-learning is not very stable and the final performance heavily depends on the initialization of the Q-function. While there are results about the asymptotic behaviour of RVIQ, at the best of our knowledge, no finite-time guarantee and/or guidance on how to tune parameters is available. Concerning RVIQ-UCB, this is a first attempt to design a model-free algorithm for the average reward. This algorithm is inspired by the recent results in finite-horizon, but further work is needed to study its regret. In the final version, we will provide more interpretation of the experiments (in particular in the continuous case).

Presentation: thanks for the comments. We will provide clearer guidance to the reader through notation and tools/results from past references.

[Meta-Review · NeurIPS 2019]

The paper considers regret minimization in infinite-horizon undiscounted MDPs using the idea of an "exploration bonus": exploring the environment by planning over an MDP with rewards that are perturbed by a bonus that scales inversely with the number of times this state-action pair was visited. Based on this idea, new online algorithms are developed, and while their regret guarantees do not improve over previous work, they are computationally efficient in contrast to existing methods. The paper has received solid support from all three reviewers, who appreciated the technical quality of the work and the advancement compared to previous work (in particular, to Bartlett & Tewari '09) in terms of computational tractability.